# LVP: LANGUAGE-GUIDE VISUAL PROJECTOR FOR EFFICIENT MULTIMODAL LLM

## ABSTRACT

Visual projector plays a crucial role in bridging the visual model and the large language model (LLM) in modern multimodal LLM. Most mllms use a simple MLP to project all visual features into visual tokens, causing a heavy computational burden and redundant visual tokens. In order to solve this problem, some efficient visual projectors, e.g., the resampler or the adaptive pooling, are developed to reduce the visual tokens. However, they only reduce the visual tokens based on the image feature, leading to the feature misalignment between visual tokens and text tokens. In this paper, we present a novel Language-guidance Visual Projector (LVP), where the text feature serves as a guide to selecting the important visual tokens. Specially, we first adopt a lightweight text encoder to extract the text feature. Then, a lightweight cross-modal feature enhancement module is proposed to enhance the cross-modal feature alignment. Finally, we select the important visual tokens according to the feature similarity between visual tokens and text tokens and apply a deformable attention module to integrate the visual features from the visual encoder into the selected visual tokens. We further propose a multi-level language-guidance visual projector, which selects the visual tokens from different stages of the visual encoder. Extensive experiments demonstrate that our LVP compresses the visual tokens by more than 75% while achieving the best performance among the existing visual projectors. For instance, LLaVA1.5-LVP with Qwen2.5-14B obtains 72.4% accuracy on VQA[T], realizing the state-of-the-art result. The code and the model will be released.

## 1 INTRODUCTION

Large Language Models (LLMs) (Touvron et al., 2023b;a; Achiam et al., 2023; Bai et al., 2023a) have made significant progress in recent years, promoting the rapid development of Multimodal Large Language Models (MLLMs). The main idea for MLLMs is to employ a visual projector to bridge the visual model and the LLM and train the visual projector using multimodel data while keeping the parameters of the visual model and LLM. Such a simple paradigm enables MLLMs to preserve and utilize the pre-training knowledge of visual model and LLM, making MLLMs show a strong capability in vision-language reasoning (Liu et al., 2024b), understanding (Alayrac et al., 2022), and interaction capabilities (You et al., 2023).

The efficiency of MLLM gains more and more attention due to the limit of compute resource in the practical use. The recent works (Li et al., 2023c; 2024c) show that LLM dominates the major computational resource and the number of input tokens directly affects the efficiency of LLM. Meanwhile, the number of visual tokens is much more than the number of text tokens in MLLMs. Reducing the number of visual tokens outputted by the visual projector is an effective way to improve efficiency. Besides, the quality of visual tokens affects the overall performance of MLLMs. Therefore, a visual projector, generating fewer but better visual tokens, is important for efficient MLLM.

Current research on the visual projector can be summarized into two lines: learnable query-based and linear projector-based. As for the learnable query-based methods, Q-Former (Li et al., 2023c) and resampler (Bai et al., 2023b) are the typical work. Both of them utilize learnable queries to squeeze and extract the visual features. However, DeCo (Yao et al., 2024a) demonstrates the training efficiency of the resampler and Q-former is low when training data is limited. As for the linear projector-based methods, such as MLP, they map the visual contexts into visual tokens without squeezing the visual

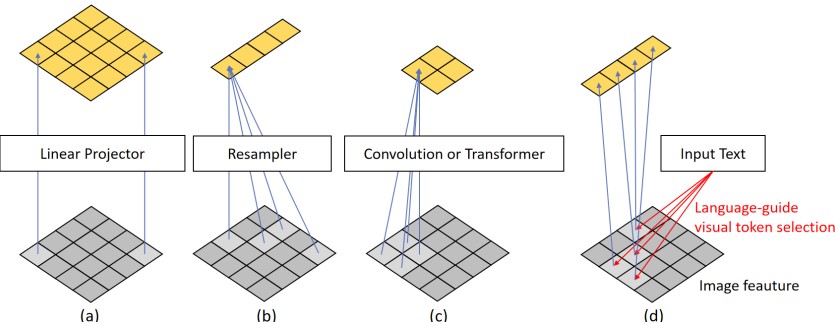

Figure 1: Visual projector comparison. (a) Linear projector, e.g., MLP. (b) Resampler. (c) Convention-based or transformer-based projector such as LDP and TokenPacker. (d) Our LVP. LVP adopts the language knowledge to select the important visual tokens, but existing visual projectors only depend on image features to reduce the visual tokens.

features. Nevertheless, this way generates numerous visual tokens, leading to a heavy computational burden. In order to squeeze the visual features while keeping visual information. Recent studies, e.g. LDP (Chu et al., 2023), Abstract (Cha et al., 2024), and DeCo (Yao et al., 2024a), use the convolution or average pooling to reduce the visual tokens and enhance the local feature. These methods inevitably lose the finer detailed features. Mini-Gemini (Li et al., 2024d) and TokenPacker (Li et al., 2024c) adopt the transformer or the cross-attention module to enrich the detailed visual information. As depicted in Figure 1, existing visual projectors focus on generating representative visual tokens only by the image feature, ignoring that inputting the visual tokens aligned with text tokens into LLM can help MLLM learn multimodal features better.

In this paper, we present a novel visual projector, named Language-guide Visual Projector (LVP). The main idea for LVP is utilizing the text feature as an guidance to decide which visual tokens should be input into LLM. Specifically, LVP employs a lightweight text encoder to extract the text feature. Then, we design a cross-modal feature enhancement module, including image-to-text and text-to-image attention, to improve the cross-modal feature alignment. Finally, LVP uses the text feature to select the important visual tokens and applies a deformable attention module to integrate the key visual features into the selected visual tokens. Such the visual token selection guided by text feature not only aligns the visual tokens with text tokens but also prompts the computational efficiency during the text generation phase of MLLM. Furthermore, to obtain fine-grained visual features, we propose a multi-level language-guide visual projector to select important visual tokens from different stages of the visual encoder. Extensive experiments are conducted across various multimodal benchmarks to evaluate the effectiveness of our approach. Notably, LLaVA1.5 with our LVP only uses 25% visual tokens (144 vs. 576) and achieves state-of-the-art performance (see Table 1).

Our main contributions are summarized as follows: **1)** We present a novel Language-guide Visual Projector (LVP) to select the visual tokens by the text feature, effectively aligning the visual tokens and text tokens. To the best of our knowledge, we are the first to adopt the language knowledge to reduce the number of visual tokens. **2)** We further propose a multi-level language-guide visual projector to generate the visual tokens from different stages of the encoder, which can capture fine-grained and global features at the same time. **3)** Experimental results demonstrate that LVP significantly reduces the visual tokens and obtains consistent performance improvement.

## 2 RELATED WORK

### 2.1 MULTIMODAL LARGE LANGUAGE MODELS

Early efforts (Li et al., 2021; Tan & Bansal, 2019) construct a series of architectures for Multimodal Large Language Models (MLLMs), consisting of a visual encoder and a language model. With the rapid development of LLM (Touvron et al., 2023b;a; Bai et al., 2023a; Achiam et al., 2023; Bi et al., 2024), many studies (Li et al., 2024a; Bai et al., 2023b; Chen et al., 2023a) focus on infusing visual features into LLM with a visual projector. LLaVA (Li et al., 2024a) feeds all visual tokens into

LLM and trains the model via visual instruction tuning, enabling the LLM to comprehend the image features and generate the correct response. MobileVLMV2 (Chu et al., 2024) proposes a 1B/3B model to benefit the resource-constrained scenarios. Qwen-VL (Bai et al., 2023b) pretrains the model with a large-scale dataset, effectively scaling up the MLLMs. Recent MLLMs, e.g., InternVL (Chen et al., 2024b) and MiniCPMV (Yao et al., 2024b), adopt an effective visual projector to enhance the model efficiency, indicating the visual projector is a significant topic to be investigated.

## 2.2 VISUAL PROJECTOR IN MLLMS

Modern MLLMs adopt the visual projector to connect the visual encoder and LLM. Early works, such as the linear projector in LLaVA (Li et al., 2024a) and MiniGPT2 (Chen et al., 2023a), preserve all visual features and map them into the language space via the fully connected layer. This approach significantly increases the computational burden due to the generation of numerous visual tokens. To reduce training resources, some efficient visual projectors have been proposed. Q-former (Li et al., 2023c) and resampler (Bai et al., 2023b) utilize a group of learnable queries to squeeze the visual features. Although such a learnable architecture reduces training resources, it underperforms in scenarios with limited training data. An alternative research direction uses convolution or pooling to reduce visual tokens. Abstractor (Cha et al., 2024) and LDP (Chu et al., 2023) leverage convolution layers to extract visual features and output compressed visual tokens. DeCo (Yao et al., 2024a) demonstrates the adaptive average pooling layer is an efficient way to compress the visual token. However, these methods neglect the fine-grained information, hurting the visual reasoning capabilities of MLLMs. TokenPacker (Li et al., 2024c) and MiniGemini (Li et al., 2024d) address this by employing cross-attention layers to inject fine-grained information from high-resolution images into compressed visual tokens. Nevertheless, their approaches focus on local regions, overlooking the global information, leading to suboptimal performance in learning global semantic features. Other approaches, such as Pixel-Shuffle (Chen et al., 2024a) and nearby concatenation (Dong et al., 2024b), directly permute the length dimension and the channel dimension, distorting intrinsic characteristics. In contrast to the existing methodologies, our LVP treats the text feature as an effective guide to select the important visual tokens.

## 3 METHOD

### 3.1 OVERVIEW

The goal of the Multimodal Large Language Model (MLLM) is to generate the response corresponding to the input instruction. In this paper, MLLM receives the image and text (instruction) as the inputs and outputs the text (response) in an autoregressive manner. Formally, the multimodal input token consists of two types: image token $X_{img}$ and text token $X_{text}$. Then, the large language model (LLM) generates the response $\mathbf{Y} = \{g_i\}_{i=1}^L$ conditioned on the $X_{img}$ and $X_{text}$, where $L$ is the number of tokens in the response. The process of multimodal generation can be formulated by

$$p(\mathbf{Y}|X_{img}, X_{text}) = \prod_{i=1}^{L} p(g_i|X_{img}, X_{text}, g_{<i}). \tag{1}$$

where $p$ denotes the conditional probability.

**Model architecture**. The architecture of MLLM is composed of three parts: visual encoder, visual projector, and LLM. The visual encoder outputs a sequence of image features. The visual projector translates the image features into a sequence of image tokens that LLM can interpret. LLM processes the text token and image token and generates the response autoregressively. In MLLM, the efficiency is mainly affected by the number of visual tokens fed into LLM. To improve the efficiency of MLLM, effective visual projectors are developed to reduce the visual tokens.

### 3.2 MOTIVATION

The role of the visual projector is to bridge the visual encoder and LLM. As described in (Li et al., 2023c; Cha et al., 2024; Li et al., 2024c), the number of visual tokens affects the overall efficiency of MLLM. Considering the scenarios of processing multiple images and large images, numerous

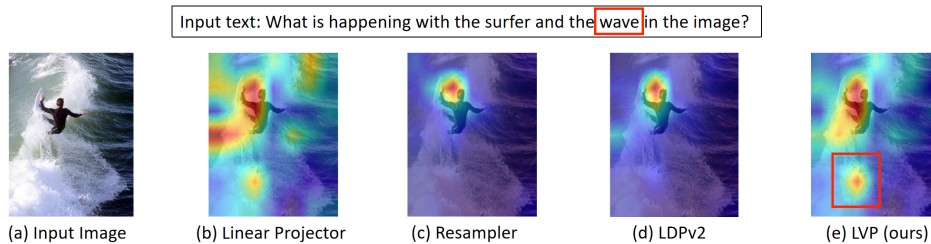

Figure 2: Comparison of attention map of different visual projectors. We visualize the attention map of input visual tokens of LLM. The implementation of attention map visualization is presented in Appendix A.1.

visual tokens are unbearable for MLLM. Improving the efficiency and scalability of MLLM is highly required. This requirement makes recent MLLM (Bai et al., 2023b; Zhu et al., 2023; Xue et al., 2024) prefer to adopt the resampler or convolution-based projector instead of the linear projector.

As shown in Figure 1, existing methods reduce the visual tokens only depending on the image feature. However, we argue that visual tokens fed into LLM should align with the text tokens. To verify this point, we visualize the attention map of visual tokens outputted by different visual projectors in Figure 2. We can observe that resampler (Li et al., 2023c) and LDPv2 (Chu et al., 2023) only focus on the feature of the surfer, ignoring the feature of the wave. From the attention map of the linear projector, we can see that the features of both wave and surfer should be considered. The reason can be attributed to the fact that the pre-training task of the visual encoder usually focuses on learning the features of foreground objects in the image, such as the surfer in this picture, but this causes the visual projector to ignore the important background information contained in the text (instruction), resulting in misalignment between the visual tokens and the text tokens.

Stemming from the above analysis, we propose a novel Language-guide Visual Projector (LVP) to align the visual tokens and text tokens. LVP follows two key principles: 1) effective alignment between visual tokens and text tokens. 2) flexibility over the number of visual tokens. Our LVP can not only determine the number of the visual tokens fed into LLM flexibly to improve the computational efficiency but also boost the overall performance of MLLM by aligning the visual tokens and text tokens.

### 3.3 LANGUAGE-GUIDE VISUAL PROJECTOR

**Architecture**. The overall architecture of our MLLM is shown in Figure 3. LVP consists of three parts: text encoder, cross-modal feature enhancement, and language-guide visual token selection. Specifically, given an input text (instruction) and an image, the visual encoder outputs the image feature $X_I \in R^{N_I \times D}$, where $N_I$ and $D$ denote the number of image tokens and dimension of $X_I$. Text encoder is composed of two transformer layers and each transformer layer contains a self-attention layer and a feed-forward network (FFN). Experimental results (see Table A1 in the Appendix) demonstrate that such a lightweight text encoder is enough for text feature extraction.

**Cross-modal feature enhancement**. Inspired by GLIP (Li et al., 2022), LXMERT (Tan & Bansal, 2019), and Grounding-DINO (Liu et al., 2023b), we introduce a lightweight Cross-modal Feature Enhancement module (CFE) to prompt the efficiency of cross-modal feature learning. CFE includes an image-to-text attention module and a text-to-image attention module. As depicted in Figure 3, the process of the image-to-text attention module can be formulated by $X_{to} = Attention(X_I, X_T, X_T)$, where $X_{to}$ is the enhanced text feature, $X_T \in R^{N_T \times D}$ is the text feature from the text encoder, $N_T$ is the number of text tokens, and $Attention(Query, Key, Value)$ represents a standard cross-attention module. In the same way, the process of the text-to-image attention module can be expressed by $X_{io} = Attention(X_{to}, X_I, X_I)$, where $X_{io} \in R^{N_I \times D}$ stands for the enhanced image feature. LXMERT adopts a similar cross-modal encoder to enhance cross-modal feature learning. Our CFE differs from it in two aspects: 1) Image-to-text attention and Text-to-image attention in LXMERT are parallel, while our CFE is a sequential structure. 2) The cross-modality encoder in the LXMERT structure is much heavier than CFE. Experimental results (see Table A4 in the Appendix) show

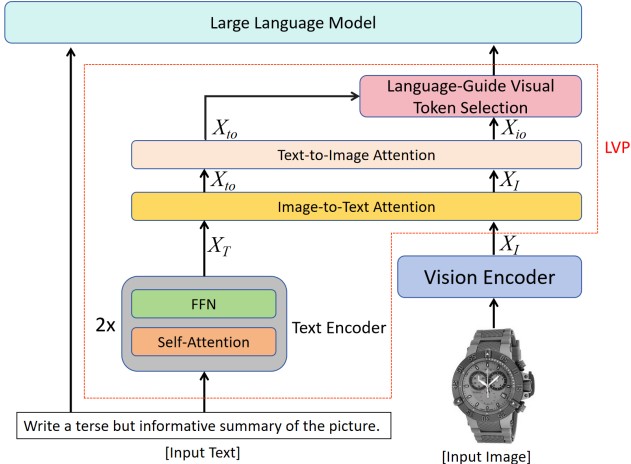

Figure 3: The overall framework of the MLLM with our LVP as the visual projector. LVP consists of three components: the lightweight text encoder to extract the text feature, the cross-modal feature enhancement module, including image-to-text attention and text-to-image attention, to enhance the cross-modal feature, and the language-guide visual token selection to reduce the visual tokens.

that such a lightweight structure is enough for our metohd, since LLM is mainly responsible for cross-modal feature interaction in MLLM.

**Language-guide visual token selection**. Our Language-guide visual token selection contains two components: visual token selection and a deformable attention module. The process of visual token selection is

$$M_{N_q} = \text{Top}_{N_q}(\text{Max}^{-1}(\frac{X_{io}X_{to}^T}{\|X_{io}\|\|X_{to}^T\|})). \tag{2}$$

where $\text{Top}_{N_q}$ denotes the operation to select the top $N_q$ visual tokens. The operation $\text{Max}^{-1}$ represents the Max operation along the $-1$ dimension, $\|\cdot\|$ is the L2 norm, and the symbol $T$ stands for the matrix transposition. Directly inputting the selected visual tokens causes the loss of visual features, we adopt a deformable attention module to enrich the feature representation of the selected visual tokens. Specifically, we take the selected visual tokens $M_{N_q}$ as the query and $X_{io}$ as the key and value. Then we input them into a deformable attention module (Zhu et al., 2020) to integrate the key visual features into the selected visual tokens. This process can be formulated by $X_{img} = DeformAttn(X_{io}, M_{N_q}, M_{N_q})$. Here, $DeformAttn(Query, Key, Value)$ denotes the deformable attention.

**Comparison with token selection in Grounding-DINO**. Grounding-DINO adopts a similar language-guidance token selection module to determine the number of object queries. Our method differs from is in two folds: 1) our language-guidance visual projector is much lighter than that in Grounding-DINO; 2) we employ the deformable attention to integrate the key visual feature into visual tokens but Grounding-DINO adopts a heavy cross-modal decoder to achieve feature interaction. Experimental results (see Table 7) show that such a simple and lightweight module achieves a similar performance compared to the heavy structure in Grounding-DINO.

## 3.4 MULTI-LEVEL LANGUAGE-GUIDANCE VISUAL PROJECTOR

To further improve the performance of MLLM, we propose a multi-level language-guide visual projector. Visual features from different stages of the visual encoder represent different visual information, e.g., visual features from the shallow stage contain rich detailed features while visual features from the deep stage tend to represent the global semantic feature. Specifically, we first divide the layers of the visual encoder into four stages following TokenPacker (Li et al., 2024c). Then, for each stage, we select the top $N_a$ visual tokens as Eq. 2. The total number of visual tokens fed into LLM is $N_a \times 4$. Finally, all selected visual tokens are concatenated along the feature dimension. In this way, the visual tokens include both detailed features and global semantic features. The overall pipeline of the multi-level language-guide visual projector is shown in Figure 4.

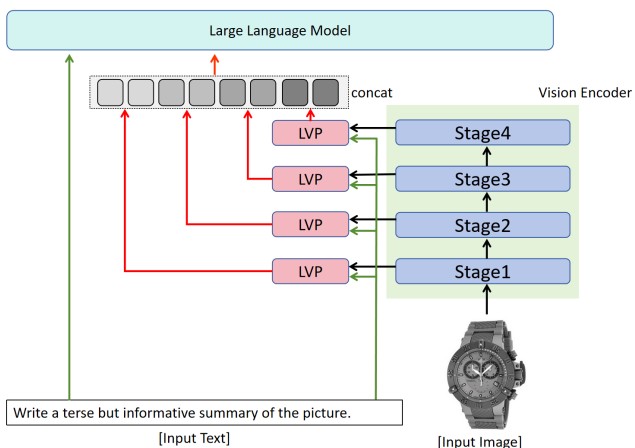

Figure 4: The pipeline of multi-level language-guide visual projector.

## 4 EXPERIMENTS

### 4.1 DATASETS

We evaluate our language-guide visual projector under the normal resolution and high resolution settings. The training process is divided into two stages. For the normal resolution, we train our model on LAION-CC-SBU-558K to achieve modal alignment in the first stage. In the second stage, we utilize 656K mixture dataset for visual instruction tuning. For the high resolution setting, we employ 1.2M training samples for the first stage and 1.5M training samples for the second stage, following Mini-Gemini (Li et al., 2024d). The evaluation dataset is composed of: $VQA^{v2}$ (Goyal et al., 2017), GQA (Hudson & Manning, 2019), VizWiz (Gurari et al., 2018) for General visual question answering; TextVQA ($VQA^T$) (Singh et al., 2019), OCRBench (OCRB) (Liu et al., 2023d), and DocmentVQA (DocVQA) (Mathew et al., 2021) for the OCR task; 3. POPE (Li et al., 2023d) for the Hallucination; 4. MMBench (MMB) (Liu et al., 2023c), MM-Vet (Yu et al., 2023), and MMMU (Yue et al., 2024).

In order to further evaluate the effectiveness of our method, we conduct the experiments in the scenario of multi-round conversations and video. For the multi-round conversations, we train our model on MMDU-45K (Liu et al., 2024c), containing 45K high-quality conversation data for the training and 110 multi-turn dialogues with more than 1600 questions for the test. Following LLaVA-OneVision (OV) (Li et al., 2024a), we adopt 4.6M high-quality knowledge data and 4.8M visual instruction data for the training. We evaluate the video performance of LVP on ActivityNet-QA (Yu et al., 2019), EgoSchema (Mangalam et al., 2023), MLVU (Zhou et al., 2024), MVBench (Li et al., 2024b), NextQA (Xiao et al., 2021), PerceptionTest (Patraucean et al., 2024), SeedBench (Li et al., 2023b), VideoChatGPT (Maaz et al., 2023), VideoDetailCaption (Li et al., 2024a), VideoMME (Fu et al., 2024), and LoneVideoBench (Wu et al., 2024).

### 4.2 IMPLEMENTATION DETAILS

In this paper, we adopt CLIP-ViT-L/14-336px (Radford et al., 2021) as the image encoder with $336 \times 336$ resolution and employ Vicuna-7B/13B (Zheng et al., 2024) as the LLM. Following LLaVA1.5, we train the model in two stages, i.e., the first stage for pretraining and the second stage for visual instruction tuning. The image encoder is frozen during the training. The number of layers of four stages in the multi-level language-guide visual projector are 12, 16, 22, and 23, respectively. We initialize the weight of the text enncoder using the first two layers of Bert (Devlin, 2018) and adopt the tokenizer of bert as the tokenizer of text encoder. We train the model for one epoch and all experiments are conducted on 8 Ascend 910B GPUs with 65 GB memory.

### 4.3 COMPARISON WITH STATE-OF-THE-ART METHODS

**Normal Resolution**. We first perform the comparison under the normal resolution setting. As shown in Table 1, in the OCR-related benchmarks (e.g., $VQA^T$, OCRB, and DocVQA), our LVP achieves better performance than the peers. For example, in DocVQA, LLaVA-LVP utilizes only 25% (144

Table 1: Comparison with state-of-the-art methods on zero-shot benchmarks. Our LVP compresses the visual tokens from 576 to 144, 64, or 36 following TokenPacker (Li et al., 2024c). * denotes reproduction results on Ascend 910B and $\sharp$ represents the multi-level language-guide visual projector.

| Method | LLM | Res. | #Token | TPS | VQA$^T$ | OCRB | DocVQA | MMB | MMMU | MME | MM-Vet | VQA$^{v2}$ | VizWiz | GQA | POPE |
|---|---|---|---|---|---|---|---|---|---|---|---|---|---|---|---|
| MobileVLM V2 (Chu et al., 2024) | MLLaMA-2.7B | 336 | 144 | 26.7 | 57.5 | – | – | 57.7 | – | 1441/- | – | – | – | 61.1 | 84.7 |
| Shikra (Chen et al., 2023b) | Vicuna-13B | 224 | 256 | 2.7 | – | – | – | 58.8 | – | – | – | 77.4 | – | – | – |
| Qwen-VL Bai et al. (2023b) | Qwen-7B | 448 | 256 | 12.5 | – | – | 65.1 | 38.2 | – | – | – | 78.8 | 35.2 | 59.3 | – |
| TokenPacker (Li et al., 2024c) | Vicuna-7B | 336 | 144 | 25.4 | 56.9 | 286 | 59.2 | 65.1 | 31.7 | 1478/- | 33.0 | 77.9 | 52.0 | 61.9 | 87.0 |
| DeCo (Yao et al., 2024a) | Vicuna-7B | 336 | 144 | 28.3 | 56.2 | – | – | – | – | 1373/- | – | 74.0 | 49.7 | 54.1 | 85.9 |
| Qwen-VL-Chat Bai et al. (2023b) | Qwen-7B | 448 | 256 | 12.5 | – | – | 62.6 | 60.6 | – | 1488/- | – | 78.2 | 38.9 | 57.5 | – |
| LLaVA1.5 (Liu et al., 2023a)* | Vicuna-7B | 336 | 576 | 4.9 | 57.3 | 291 | 58.7 | **67.7** | 30.3 | 1370/294 | 32.2 | 78.4 | 50.0 | 62.0 | 87.3 |
| LLaVA1.5-LVP$^\sharp$ | Vicuna-7B | 336 | 144 | 24.2 | **58.9** | 317 | 59.7 | 67.3 | 30.6 | **1495/304** | **34.5** | **79.2** | **53.1** | **62.5** | **88.0** |
| LLaVA1.5 (Liu et al., 2023a)* | Vicuna-13B | 336 | 576 | 1.8 | 59.7 | 320 | 60.0 | 68.3 | 31.0 | 1475/310 | **36.5** | 81.4 | 54.9 | 64.3 | 87.0 |
| LLaVA1.5-LVP$^\sharp$ | Vicuna-13B | 336 | 144 | 8.3 | **60.0** | 327 | 60.5 | **68.6** | 31.5 | **1480/305** | 35.3 | **81.6** | **56.2** | **65.2** | **87.9** |
| **Fewer Tokens Setting** | | | | | | | | | | | | | | | |
| InstructBLIP (Dai et al., 2023) | Vicuna-7B | 224 | 64 | 28.8 | 50.1 | – | – | 36.0 | – | – | 26.2 | – | 34.5 | 49.2 | – |
| InstructBLIP (Dai et al., 2023) | Vicuna-13B | 224 | 64 | 12.9 | 50.7 | – | – | – | – | – | 25.6 | – | 33.4 | 49.5 | – |
| TokenPacker (Li et al., 2024c) | Vicuna-7B | 336 | 64 | 25.3 | 55.4 | 269 | 58.0 | 64.1 | 30.5 | 1435/- | 31.7 | 77.2 | 50.7 | 61.1 | 86.3 |
| TokenPacker (Li et al., 2024c) | Vicuna-13B | 336 | 64 | 11.7 | 57.2 | 292 | **59.5** | 66.2 | **32.0** | 1500/- | 34.2 | 78.1 | 52.9 | 62.0 | 87.3 |
| LLaVA1.5-LVP$^\sharp$ | Vicuna-7B | 336 | 64 | 24.9 | 56.0 | 275 | 58.2 | 65.7 | 30.2 | 1452/300 | 32.9 | 77.9 | 52.2 | 61.8 | 87.2 |
| LLaVA1.5-LVP$^\sharp$ | Vicuna-7B | 336 | 64 | 24.9 | **57.8** | 306 | 59.0 | **67.0** | 31.4 | **1477/303** | 34.4 | **79.2** | **53.8** | **63.6** | **87.5** |
| LLaVA-PruMerge (Shang et al., 2024) | Vicuna-7B | 336 | 32 | 38.8 | 56.0 | – | – | 60.9 | – | 1350/– | – | 72.0 | – | – | 76.3 |
| LLaVA-PruMerge (Shang et al., 2024) | Vicuna-13B | 336 | 32 | 16.7 | 58.4 | – | – | 62.3 | – | 1428/– | – | 72.8 | – | – | 78.5 |
| TokenPacker (Li et al., 2024c) | Vicuna-7B | 336 | 36 | 39.0 | 53.7 | 249 | 56.3 | 62.8 | 28.9 | 1377/– | 29.6 | 75.0 | 50.2 | 59.6 | 86.2 |
| TokenPacker (Li et al., 2024c) | Vicuna-13B | 336 | 36 | 16.4 | 57.0 | 284 | 58.6 | 66.2 | 31.5 | 1446/– | 34.1 | 76.3 | **53.9** | 60.7 | 86.5 |
| LLaVA1.5-LVP$^\sharp$ | Vicuna-7B | 336 | 36 | 36.4 | 54.0 | 255 | 57.0 | 63.6 | 29.4 | 1400/290 | 31.0 | 75.9 | 51.6 | 60.6 | 86.5 |
| LLaVA1.5-LVP$^\sharp$ | Vicuna-13B | 336 | 36 | 15.8 | **57.8** | 298 | 59.3 | **66.9** | 31.4 | **1473/299** | 34.3 | 78.7 | 53.5 | **61.8** | **87.4** |

vs. 576) visual tokens but improves the performance by 1% (59.7% vs. 58.7%) and 0.5% (60.5% vs. 60.0%) compared to the vanilla LLaVA. Compared with the latest method DeCo (Yao et al., 2024a) and TokenPacker (Li et al., 2024c), our LVP achieves 2.7% (58.9% vs. 56.2%) and 2% (58.9% vs. 56.9%) improvements on VQA$^T$, demonstrating the effectiveness of our LVP. LLaVA-LVP also achieves the promising results on the comprehensive benchmarks. For instance, LLaVA-LVP-7B gains the performance improvements by 2.3% (34.5% vs. 32.2%) on MM-Vet 3.1% (53.1% vs. 50.0%) on VizWiz, 2.5% (62.5% vs. 62.0%) on GQA, and 0.7% (88.0% vs. 87.3%) on POPE compared to vanilla LLaVA-7B. As for the 13B model, LLaVA-LVP obtains the following improvements against LLaVA: 0.3% (68.6% vs. 68.3%) on MMB, 1.3% (56.2% vs. 54.9%) on VizWiz, 0.2% (81.6% vs. 81.4%) on VQA$^{v2}$, 0.9% (65.2% vs. 64.3%) on GQA, 0.9% (87.9% vs. 87.0%) on POPE. The reason for the above results is the visual tokens outputted by the linear projector in vanilla LLaVA1.5 are redundant, causing inefficient learning on important visual features. Our LLaVA-LVP directly inputs the important visual tokens aligned with the text tokens into LLM, which can naturally improve learning efficiency. Moreover, LLaVA1.5-LVP surpasses the previous methods, e.g., Qwen-VL and DeCo. LLaVA1.5-LVP exceeds the Qwen-VL-Chat on four benchmarks with fewer visual tokens and each benckmark all gains over 2% performance improvement. Compared to the recent method DeCo, LLaVA1.5-LVP displays significant performance advantages. For instance, LLaVA1.5-LVP enhances the performance metrics by 3.4% (53.1% vs. 49.7%) on VizWiz and 5% (62.5% vs. 57.5%) on GQA. It should be noted that Qwen-VL and DeCo utilize more training data than LLaVA1.5-LVP.

**Fewer visual tokens comparison**. To further verify the effectiveness of our method, we compare LLaVA1.5-LVP with the previous leading methods under the fewer visual tokens setting. Results are shown in Table 1. Our LLaVA1.5-LVP achieves the best performance across all benchmarks. For example, for the 7B model, we achieve better performance by a large margin than the TokenPacker, which is the latest leading method, on MMB (65.7% vs. 64.1%) and VizWiz (52.5% vs. 50.7%) datasets with 64 visual tokens. When adopting 36 visual tokens, LLaVA1.5-LVP-7B gets a significant performance improvement over LLaVA-PruMerge-7B on MMB (63.6% vs. 60.9%) and VQA$^{v2}$ (75.9% vs. 72.0%) datasets. the above methods all focus on selecting the important visual tokens. However, their visual token selection strategy only depends on the image feature, leading to feature misalignment between the visual tokens and text tokens. Our LLaVA1.5-LVP chooses the important visual tokens based on both image and text features, effectively aligning the tokens of two modalities. The results demonstrate that an effective visual token selection strategy should generate the visual tokens correlated to text tokens.

**High-Resolution**. We further evaluate the performance of LVP under the high-resolution setting and results are shown in Table 2. Following TokenPacker (Li et al., 2024c), we set the input resolution to 1088×1088 and 1344×1344. We compare our LVP against the latest MLLM with high resolution, including OtterHD (Li et al., 2023a), Sphinx-2k (Lin et al., 2023), Monkey (Li et al.,

Table 2: Performance comparisons with high-resolution approaches on nine benchmarks. The best results are **bold** and the second-best results are underlined. * denotes the reproduction results on Ascend 910B and ♯ represents the multi-level language-guide visual projector. ‡, ¶, and ♣ denotes the scaling factor $s = 2, 3, 4$ in TokenPacker, respectively. ∼ means approximately equal to.

| Method | LLM | Max Res. | #Token | TPS | VQA$^T$ | OCRB | DocVQA | MMB | MMMU | MME | MM-Vet | VQA$^{v2}$ | VizWiz | GQA | POPE |
|---|---|---|---|---|---|---|---|---|---|---|---|---|---|---|---|
| OtterHD (Li et al., 2023a) | Fuyu-8B | 1024×1024 | – | 0.8 | – | – | – | 58.3 | – | 1294/– | 26.3 | – | – | – | 86.0 |
| SPHINX-2k (Lin et al., 2023) | LLaMA-13B | 762×762 | 2890 | 0.4 | 61.2 | – | – | 65.9 | – | 1471/– | 40.2 | 80.7 | 44.9 | 63.1 | 87.2 |
| UReader (Ye et al., 2023) | LLaMA-13B | 896×1120 | – | 0.08 | 57.6 | – | 65.4 | – | – | – | – | – | – | – | – |
| Monkey (Li et al., 2024e) | QWen-7B | 896×1344 | 1792 | 1.1 | – | 514 | – | – | – | – | – | 80.3 | 61.2 | 60.7 | 67.6 |
| TextHawk (Yu et al., 2024) | InternLM-7B | 1344×1344 | – | 0.2 | – | – | **76.4** | 74.6 | – | 1500/- | – | – | – | 64.6 | – |
| LLaVA-UHD (Xu et al., 2024b) | Vicuna-13B | 672×1008 | – | 0.1 | 67.7 | – | – | 68.0 | – | 1535/– | – | 81.7 | 56.1 | 65.2 | **89.1** |
| LLaVA-NeXT (Liu et al., 2024a) | Vicuna-7B | 672×672 | 2880 | 0.9 | 64.9 | – | – | 67.4 | 35.8 | 1519/332 | – | 81.8 | 57.6 | – | 86.5 |
| LLaVA-NeXT (Liu et al., 2024a) | Vicuna-13B | 672×672 | 2880 | 0.5 | 67.1 | – | – | 70.0 | 36.2 | 1575/326 | – | 82.8 | 60.5 | – | 86.2 |
| Mini-Genimi-HD (Li et al., 2024d) | Vicuna-7B | 1536×1536 | 2880 | 1.0 | 68.4 | 456* | 65.0* | 65.8 | 36.8 | 1546/319 | 41.7* | 80.3* | 54.6* | – | 86.8* |
| Mini-Genimi-HD (Li et al., 2024d) | Vicuna-13B | 1536×1536 | 2880 | 0.6 | 70.2 | 501* | 70.0* | 68.6 | 37.3 | 1575/326 | 51.0* | 81.5* | 57.2* | – | 87.0* |
| TokenPacker (Li et al., 2024c) | Vicuna-7B | 1088×1088 | ~954‡ | 2.0 | 68.0 | 452 | 60.2 | 67.4 | 35.4 | 1489/338 | 42.5* | 81.2 | 54.7 | 64.8* | 88.2 |
| TokenPacker (Li et al., 2024c) | Vicuna-13B | 1088×1088 | ~954‡ | 1.3 | 69.3 | 498 | 63.0 | 69.5 | 38.8 | 1595/356 | 45.0* | 82.0 | 59.2 | 65.9* | 88.1 |
| TokenPacker (Li et al., 2024c) | Vicuna-13B | 1344×1344 | ~1393‡ | 0.9 | 70.6 | 521 | 70.0 | 68.7 | 37.4 | 1574/350 | 45.8* | 81.7 | 57.0 | 65.5* | 88.0 |
| TokenPacker (Li et al., 2024c) | Vicuna-13B | 1344×1344 | ~619¶ | 1.5 | 68.8 | 470 | 63.0 | 69.9 | 38.2 | 1577/353 | 44.2* | 81.7 | 61.0 | 64.9* | 87.6 |
| TokenPacker (Li et al., 2024c) | Vicuna-13B | 1344×1344 | ~347♣ | 2.0 | 68.4 | 447 | 58.0 | 68.3 | 36.9 | 1577/332 | 43.9* | 81.2 | 58.1 | 64.0* | 88.0 |
| LLaVA1.5-LVP♯ | Vicuna-7B | 1088×1088 | 954 | 1.9 | 68.8 | 503 | 61.0 | 68.4 | 36.2 | 1582/350 | 43.1 | 81.9 | 55.9 | 65.2 | 88.2 |
| LLaVA1.5-LVP♯ | Vicuna-13B | 1088×1088 | 954 | 1.3 | 69.7 | 519 | 64.9 | 69.9 | 39.8 | 1600/367 | 45.7 | 82.5 | 60.4 | 66.4 | 88.2 |
| LLaVA1.5-LVP♯ | Qwen2.5-7B | 1088×1088 | 954 | 2.1 | 71.3 | 527 | 68.0 | 70.3 | **40.3** | 1633/371 | 46.4 | **82.9** | 60.8 | 66.9 | 88.3 |
| LLaVA1.5-LVP♯ | Vicuna-13B | 1344×1344 | 1393 | 1.0 | 71.8 | 526 | 72.4 | 69.5 | 39.2 | 1592/367 | 46.6 | 82.2 | 60.3 | 66.7 | 88.3 |
| LLaVA1.5-LVP♯ | Vicuna-13B | 1344×1344 | 619 | 1.4 | 69.2 | 512 | 64.5 | 70.3 | 39.5 | 1595/361 | 45.2 | 82.2 | 61.0 | 66.0 | 88.1 |
| LLaVA1.5-LVP♯ | Vicuna-13B | 1344×1344 | 347 | 2.3 | 69.0 | 509 | 61.2 | 68.5 | 36.8 | 1598/349 | 44.3 | 82.0 | 59.3 | 64.6 | 88.2 |
| LLaVA1.5-LVP♯ | Qwen2.5-14B | 1344×1344 | 1393 | 1.1 | **72.4** | **533** | 73.0 | 71.5 | **40.3** | **1652/374** | **47.0** | 82.7 | **61.3** | **67.0** | 88.3 |

Table 3: Evaluation results of different methods on MMDU. We report the metrics of Creativity (C), Richness (R), Visual Perception (VP), Logical Coherence (LC), Answer Accuracy (AA), Image Relationship Understanding (IRU), and the averaged (Avg.) results. Param represents the size of LLM.

| Models | Param | C | R | VP | LC | AA | IRU | Avg. |
|---|---|---|---|---|---|---|---|---|
| LLaVa1.5-7B (Liu et al., 2023a) | 7B | 27.8 | 28.0 | 33.2 | 43.0 | 35.4 | 31.7 | 32.2 |
| Qwen-VL-7B (Bai et al., 2023b) | 7B | 33.4 | 33.6 | 39.2 | 53.8 | 43.1 | 38.1 | 39.3 |
| InternLM-XC2 (Dong et al., 2024a) | 7B | 29.7 | 29.5 | 36.2 | 50.1 | 43.4 | 35.2 | 35.6 |
| MiniCPM-v-2.5 (Yao et al., 2024b) | 8B | 27.0 | 26.4 | 33.2 | 48.9 | 38.6 | 32.2 | 33.0 |
| Deepseek-VL (Lu et al., 2024) | 8B | 27.3 | 27.7 | 31.2 | 38.7 | 33.2 | 30.0 | 30.8 |
| InternVL-Chat-V1.5 (Chen et al., 2024a) | 26B | 31.2 | 31.5 | 37.4 | 52.6 | 41.7 | 36.1 | 37.4 |
| LLaVa1.5 + MMDU-45k | 7B | 34.3 | 34.5 | 36.7 | 47.2 | 38.5 | 35.5 | 37.2 |
| LLaVA1.5-LVP + MMDU-45k | 7B | **34.7** | **35.0** | **37.8** | **49.0** | **40.0** | **36.0** | **38.8** |
| InternLM-XC2 + MMDU-45k | 7B | 45.6 | 43.9 | 49.9 | 64.1 | 53.0 | 48.7 | 50.1 |
| InternLM-XC2-LVP + MMDU-45k | 7B | **46.0** | **44.4** | **51.0** | **65.7** | **53.8** | **49.0** | **51.7** |

2024e), Texthawk (Yu et al., 2024), UReader (Ye et al., 2023), LLaVA-UHD (Xu et al., 2024b), LLaVA-Next (Liu et al., 2024a), and Mini-Gemini-HD (Li et al., 2024d). Eleven benchmarks, i.e., OCR-related VQA$^T$, OCRB, and DocVQA, and comprehensive MMB, MMMU, MME, MM-Vet, VQA$^{v2}$, VizWiz, GQA, and POPE, are utilized to perform the overall evaluation. With 619 visual tokens, our method gets the second-best performance on MMB, MMMU, and VizWiz, superior to the methods with many visual tokens (e.g. TokenPacker. Mini-Genimi-HD, and LLaVA-NeXT). For the OCR tasks, our LLaVA1.5-LVP with Qwen2.5-14B achieves state-of-the-art performance on OCR-related VQA$^T$ (72.4%). LLaVA1.5-LVP with Vicuna 13B surpasses the second-base method TokenPacker by 1.2% (71.8% vs. 70.6%). These results demonstrate that selecting the important visual tokens effectively is more meaningful than the number of visual tokens for the high-resolution setting. On the other hand, our approach obtains the best performance at a lower resolution ($\leq$ 1088×1088). The experimental results validate the effectiveness of our LLaVA1.5-LVP.

**Multi-round conversations**. We evaluate LVP in the scenario of multi-round conversations and results are shown in Table 3. InternLM-XC2-LVP establishes the new state-of-the-art results on each metric. It can be seen that LVP gains 1.6% improvement for LLaVA1.5 (38.8% vs. 37.2%) and InternLM-XC2 (51.7% vs. 50.1%). LVP improves performance by over 1% in terms of visual perception and logical coherence. The results demonstrate that LVP works for multi-round conversations.

**Video Benchmarks**. We evaluate the effectiveness of LVP under the video task. LVP improves the model performance on all 11 benchmarks, showing its advantages in the video tasks. LVP gains 1.2% (47.0% vs. 45.8%) and 1.4% (57.8% vs. 56.4%) for the 0.5B and 7B model on LongVideoBench,

Table 4: LLaVA-OneVision-LVP performance on video benchmarks. We report the score out of 5 for VideoDetailCaption (VideoDC), VideoChatGPT while other results are reported in accuracy. All results are reported as 0-shot accuracy. The number of visual tokens fed into LLM in LLaVA-OV is Z×196, where Z is the sampled frame per video. The number of visual tokens fed into LLM in LLaVA-OV-LVP is Z×98.

| Model | ActNet-QA | EgoSchema | MLVU | MVBench | NextQA | PercepTest | SeedBench | VideoChatGPT | VideoDC | VideoMME | LongVideoBench |
|---|---|---|---|---|---|---|---|---|---|---|---|
| | test | test | m-avg | test | mc | val | video | test | test | wo/w-subs | val |
| VILA-40B Lin et al. (2024) | 58.0 | 58.0 | - | - | 67.9 | 54.0 | - | 3.36 | 3.37 | 60.1/61.1 | - |
| PLLaVA-34B Xu et al. (2024a) | **60.9** | - | - | 58.1 | - | - | - | 3.48 | - | - | - |
| LLaVA-N-Video-34B Liu et al. (2024a) | 58.8 | 49.3 | - | - | 70.2 | 51.6 | - | 3.34 | 3.48 | 52.0/54.9 | 50.5 |
| IXC-2.5-7B Zhang et al. (2024) | 52.8 | - | 37.3 | **69.1** | 71.0 | 34.4 | - | 3.46 | 3.73 | 55.8/58.8 | - |
| LLaVA-N-Video-32B Liu et al. (2024a) | 54.3 | 60.9 | 65.5 | - | 77.3 | **59.4** | - | 3.59 | 3.84 | **60.2/63.0** | - |
| LLaVA-OV-0.5B | 50.5 | 26.8 | 50.3 | 45.5 | 57.2 | 49.2 | 44.2 | 3.12 | 3.55 | 44.0/43.5 | 45.8 |
| LLaVA-OV-LVP-0.5B | 51.0 | 28.0 | 51.0 | 46.3 | 57.9 | 50.3 | 44.9 | 3.55 | 3.77 | 45.9/44.7 | 47.0 |
| LLaVA-OV-7B | 56.6 | 60.1 | 64.7 | 56.7 | 79.4 | 57.1 | 56.9 | 3.51 | 3.75 | 58.2/61.5 | 56.4 |
| LLaVA-OV-LVP-7B | 57.3 | **61.0** | **65.8** | 57.8 | **80.3** | 58.3 | **57.6** | **3.70** | **3.88** | 59.9/**63.0** | **57.8** |

Table 5: Evaluation results on different visual projectors. The resolution of the input image is 336×336 and the base model is LLaVA1.5 with Vicuna-7B. We adopt token per second (TPS) to evaluate the throughput of LLM during inference, measured by a single Ascend 910B. ♯ stands for the multi-level language-guide visual projector.

| Projector | #Token | TPS | MMB | MM-Vet | VQA$^{v2}$ | GQA | POPE | VizWiz | Avg. |
|---|---|---|---|---|---|---|---|---|---|
| MLP Liu et al. (2023a) | 576 | 4.9 | 67.7 | 32.2 | 78.4 | 62.0 | 87.3 | 50.0 | 62.9 |
| Average-Pooling | 144 | **28.3** | 64.6 | 26.9 | 76.5 | 60.2 | 86.4 | 51.5 | 61.0 |
| Resampler (Bai et al., 2023b) | 144 | 24.9 | 63.1 | 28.9 | 75.3 | 58.6 | 84.8 | 52.5 | 60.5 |
| C-Abstractor (Cha et al., 2024) | 144 | 24.5 | 65.1 | 31.8 | 75.7 | 60.0 | 85.1 | 49.7 | 61.2 |
| Pixel-Shuffle (Chen et al., 2024a) | 144 | 25.6 | 64.2 | 29.6 | 76.5 | 60.6 | 85.3 | 49.2 | 60.9 |
| LDPv2 (Chu et al., 2024) | 144 | 25.5 | 65.7 | 28.9 | 77.8 | 62.1 | 86.0 | 47.9 | 61.4 |
| TokenPacker (Li et al., 2024c) | 144 | 25.4 | 65.1 | 33.0 | 77.9 | 61.8 | 87.0 | 52.0 | 62.8 |
| LVP | 144 | 25.3 | 66.2 | 33.3 | 78.5 | 62.0 | 87.8 | 52.7 | 63.4 |
| LVP♯ | 144 | 24.2 | **67.3** | **34.5** | **79.2** | **62.5** | **88.0** | **53.1** | **64.1** |
| Average-Pooling | 64 | **29.5** | 62.3 | 27.3 | 72.9 | 59.0 | 85.6 | 48.2 | 59.2 |
| Resampler (Bai et al., 2023b) | 64 | 27.2 | 63.4 | 29.5 | 74.0 | 58.0 | 83.9 | **53.2** | 60.3 |
| C-Abstractor (Cha et al., 2024) | 64 | 26.9 | 62.9 | 29.2 | 74.4 | 59.0 | 85.3 | 45.2 | 59.3 |
| Pixel-Shuffle (Chen et al., 2024a) | 64 | 28.0 | 63.4 | 28.3 | 75.0 | 59.4 | 85.0 | 47.6 | 59.7 |
| LDPv2 (Chu et al., 2024) | 64 | 27.5 | 64.0 | 30.8 | 75.2 | 60.1 | 85.8 | 49.6 | 60.9 |
| TokenPacker (Li et al., 2024c) | 64 | 25.3 | 64.1 | 31.7 | 77.2 | 61.1 | 86.3 | 50.7 | 61.9 |
| LVP | 64 | 25.7 | 64.9 | 32.3 | 77.2 | 61.4 | 86.8 | 51.4 | 62.3 |
| LVP♯ | 64 | 24.9 | **65.7** | **32.9** | **77.9** | **61.8** | **87.2** | 52.2 | **63.0** |

demonstrating its strength in long video understanding. Besides, LLaVA-OV-LVP-7B achieves better LLaVA-N-Video-32B on ActNet-QA, EgoSchema, MLVU, NextQA, VideoChatGPT, VideoDC, and LoneVideoBench, indicating that our LVP is an effective visual projector for video tasks.

## 4.4 ABLATION STUDY

In this section, we validate the effectiveness of each component of the proposed LVP. All experiments are conducted on the data as those in LLaVA1.5 and Vicuna-7B are utilized as LLM.

**Comparison of visual projectors**. We first conduct the comparison experiments between the existing visual projectors and our LVP. To analyze the inference speed, we adopt the token per second (TPS) to evaluate the throughput. We adopt the adaptive average pooling as the visual token reduction operation for the average-pooling. We just replace the MLP layers in LLaVA1.5 with the above visual projectors for a fair comparison. To analyze the inference speed, we adopt the token per second

(TPS) to measure the throughput of MLLM. From Table 5, it can be seen that our LVP achieves the best performance on all benchmarks. For example, when input visual tokens are 144, LVP without multi-level feature outperforms the latest method TokenPacker on various benchmarks, such as 1.1% (66.2% vs. 65.1%) performance improvement on MMB and 0.6% (78.5% vs. 77.9%) enhancement on VQA$^{v2}$. Compared with the convolution-based method, i.e., Average Pooling, LDPv2, and C-Abstractor, LVP shows obvious performance advantages, e.g. 2.4% (63.4% vs. 61.0%), 2.0% (63.4% vs. 61.4%), and 2.2% (63.4% vs. 61.2%) average performance improvement against Average Pooling, LDPv2, and C-Abstractor. Equipped with multi-level features, our LVP further obtains 64.1% average performance, superior to the MLP projector, which is the first visual projector that exceeds MLP. We conclude the reason why LVP surpasses MLP is that the visual tokens outputted by MLP are redundant, making the model require more training epochs to learn the important features, but our LVP selects the important visual tokens by the text feature, reducing the useless visual tokens and improving the learning efficiency. When input visual tokens are 64, our LVP with multi-level feature obtains 63.0% average performance, on par with MLP (63.0% vs. 62.9%), further indicating the effectiveness of our visual token selection approach. In terms of TPS, all visual projectors achieve significant inference speed improvement against MLP. Our LVP achieves the competitive performance compared to other visual projectors on inference speed.

**Integrating into different MLLMs**. We further integrate the proposed LVP into different MLLMs to evaluate the effectiveness of our LVP. We conduct the experiments on MiniCPMV-2.6, Qwen-VL-Chat, and MobileVLMV2 and LLM for three models are LLaMA3-8B, Qwen-7B, and Vicuna-7B, respectively. Results are shown in Table 6. We

Table 6: Results of integrating LVP into different MLLMs. The input resolution is $336\times336$.

| Method | #Token | VQA$^{v2}$ | GQA | VQA$^{T}$ | OCRB |
|---|---|---|---|---|---|
| MiniCPMV-2.6 (Yao et al., 2024b) | 144 | 83.6 | 67.3 | 58.0 | 539 |
| MiniCPMV-2.6-LVP | 144 | **84.2** | **68.9** | **58.7** | **564** |
| Qwen-VL-Chat (Bai et al., 2023b) | 144 | 78.2 | 56.6 | 52.8 | 302 |
| Qwen-VL-Chat-LVP | 144 | **79.2** | **58.3** | **53.9** | **326** |
| MobileVLMv2 (Chu et al., 2024) | 144 | 77.4 | 62.6 | 43.7 | 337 |
| MobileVLMv2-LVP | 144 | **78.7** | **62.9** | **45.0** | **353** |

can observe that our LVP achieves a consistent improvement on different MLLMs. For instance, LVP enhances the performance by 0.6%, 1.6%, 0.7%, and 25 on VQA$^{v2}$, GQA, VQA$^{T}$, and OCRB for the latest MiniCPMV-2.6. The results manifest that LVP can be a versatile visual projector to reduce the visual tokens while improving the model performance.

**Comparison with the peer in Grounding-DINO**. We compare our LVP against the peer in Grounding-DINO and adopt Vicuna-7B as LLM to perform the experiment. From Table 7, it can be seen that LVP achieves competitive performance on different benchmarks when compared with the visual projector in Grounding-DINO. However, LVP gains much faster TPS than the visual projector in Grounding-DINO. When applying the

Table 7: Comparison between the peer in Grounding-DINO and LVP. The input resolution is $336\times336$ and the number of visual tokens fed into LLM is 144. $\sharp$ represents the multi-level language-guide visual projector.

| Method | TPS | VQA$^{v2}$ | GQA | VQA$^{T}$ | OCRB |
|---|---|---|---|---|---|
| Grounding-DINO (Liu et al., 2023b) | 18.7 | 78.3 | **62.0** | 57.7 | **300** |
| LVP | **25.3** | 78.5 | **62.0** | 58.0 | 298 |
| Grounding-DINO$^{\sharp}$ (Liu et al., 2023b) | 12.1 | 79.2 | **63.1** | 59.2 | 314 |
| LVP$^{\sharp}$ | **24.2** | 79.2 | 62.5 | 58.9 | **317** |

multi-level feature, the gap between two visual projectors in TPS is further widened, demonstrating that our LVP is better than the visual projector in Grounding-DINO for efficient MLLM. Reasons we conclude may be that: 1) the deformable attention module integrates the key feature into the visual tokens, which significantly reduces the redundant feature aggregation as that in the Grounding-DINO. 2) LLM are mainly responsible for the feature interaction between visual features and text features in MLLMs, weakening the role of the heavy cross-modal decoder in Grounding-DINO.

## 5 CONCLUSION

We introduce a novel Language-guide Visual Projector (LVP) for efficient MLLM. LVP adopts the text (instruction) feature as the guidance to select the important visual tokens, effectively reducing the visual tokens while aligning the visual tokens fed into LLM with text tokens. To make full use of the features from the different stages of the visual encoder, we further propose a novel multi-level language-guide visual projector. Experimental results show that LVP achieves state-of-the-art performance among existing visual projectors. Notably, InternLM-XC2-LVP establishes the best performance on MMDU benchmark with much fewer visual tokens.

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

# A APPENDIX

## A.1 ADDITIONAL IMPLEMENTATION DETAILS

**Implementation of attention map visualization**. In this section, we describe the implementation of attention map visualization in detail. We adopt an approach similar to R-GAE in DeCo (Yao et al., 2024a). Specifically, we first construct a Text-to-Visual map $M_t \in R^{N_I \times N_q}$. $M_t$ is initialized to an identity matrix. For each layer in the projector, an attention map is obtained by utilizing the gradients to average across the attention heads for the resampler (Bai et al., 2023b) and LVP. For Linear projector and LDPv2 (Chu et al., 2024), an attention map is obtained by adopting the gradients of each layer. For generation time step $t$, we can propagate the $M_t$ from the projector's first layer to its last layer. Finally, we average the step $t$ and average the $M_t$ to get the final attention map.

**Implementation of high-resolution**. We take the high-resolution image processing in LLaVA-HD (Liu et al., 2023a) as our high-resolution image processing method. Given a high-resolution image, LLaVA-HD first splits the image into different patches and each patch is fed into the visual encoder. The visual encoder outputs a sequence of visual tokens. We use $P_{i=1}^N$ to represent the sequence of visual tokens and $N$ is the number of patches. Besides, LLaVA-HD resizes the original high-resolution image to the size the visual encoder can process. Here we use $P_H$ the denote the the visual token of the high-resolution image. Finally, LLaVA-HD concatenate the $P_{i=1}^N$ and $P_H$. We use $P_C$ to stand for the concatenated visual tokens. $P_C$ is the visual input of our LVP. We use the text feature as a guide to select the Top $N_q$ ($N_q$ is much smaller than the number of $P_C$) visual tokens from $P_C$ based on the similarity between visual features and text features.

## A.2 ADDITIONAL ABLATION STUDY

In this section, we conduct additional ablation studies to validate the effectiveness of the component of LVP. All experiments are performed as those in LLaVA1.5 with Vicuna-7B as LLM.

**Size of the text encoder**. We compare our lightweight text encoder with bert-base (Devlin, 2018) and the results are in Table A1. From Table A1, our LVP obtains a significant TPS advantage over bert-base while achieving competitive performance against bert-base. The reason may be that LVP is responsible for selecting important visual tokens not extracting text features. Therefore, adopting a heavy text encoder does not bring obvious improvement.

Table A1: Comparison between bert-base and our lightweight text encoder.

| Method | TPS | VQA$^{v2}$ | GQA | VQA$^T$ | OCRB |
|---|---|---|---|---|---|
| LVP-Bert (Devlin, 2018) | 11.5 | 78.3 | **62.2** | 57.9 | **314** |
| LVP | **25.3** | **78.5** | 62.0 | **58.0** | 298 |
| LVP$^\sharp$-Bert (Devlin, 2018) | 9.9 | **79.5** | 62.4 | **59.3** | 330 |
| LVP$^\sharp$ | **24.2** | 79.2 | **62.5** | 58.9 | 317 |

**Influence of the deformable attention module**. Table A2 demonstrates the effectiveness of the deformable attention module in LVP. We can observe that the deformable attention module brings consistent performance improvement. The results show that compressing the visual features into selected visual tokens is a necessary step for an effective visual projector, which can avoid the loss of visual features.

Table A2: Influence of the deformable attention module. DF denotes the deformable attention module and RA represents the regular attention.

| Method | TPS | VQA$^{v2}$ | GQA | VQA$^T$ | OCRB |
|---|---|---|---|---|---|
| LVP w/o DF | 26.0 | 75.4 | 59.5 | 55.7 | 269 |
| LVP w RA | 25.0 | 78.0 | 61.7 | 57.3 | 291 |
| LVP w DF | 25.3 | **78.5** | **62.0** | **58.0** | **298** |
| LVP$^\sharp$ w/o DF | 25.0 | 77.4 | 60.3 | 56.1 | 275 |
| LVP w RA | 23.9 | 78.8 | 62.0 | 58.2 | 315 |
| LVP$^\sharp$ w DF | 24.2 | **79.2** | **62.5** | **58.9** | **317** |

**Influence of the size of the cross-modal feature enhancement module**. We further ablate the size of the cross-modal feature enhancement module. Here, the size denotes the number of blocks in the cross-modal feature enhancement module. We treat the combination of image-to-text attention and text-to-image attention as a block. Results are shown in Table A3. As the number of blocks increases, the model performance is not improved significantly. For instance, the performance of $N_L = 1$ is similar to that of $N_L = 6$ (the setting

Table A3: Influence of the size of the cross-modal feature enhancement module. $N_L$ represents the number of blocks in the cross-modal feature enhancement module.

| Method | $N_L$ | VQA$^{v2}$ | GQA | VQA$^T$ | OCRB |
|---|---|---|---|---|---|
| LVP | 1 | 78.5 | 62.0 | 58.0 | 298 |
| LVP | 2 | 78.3 | 62.1 | 58.0 | 296 |
| LVP | 4 | 78.6 | 62.0 | 58.2 | 301 |
| LVP | 6 | 78.7 | 62.1 | 57.8 | 302 |

in Grounding-DINO). However, the TPS of $N_L = 1$ and $N_L = 6$ are 25.3 and 19.4, respectively. Therefore, we set $N_L$ to 1 considering the performance and TPS.

**Comparison with the peer in LXMERT.** We compare the cross-modal feature enhancement module (CFE) with the peer in LXMERT. From Table A4, we can observe that our CFE achieves the similar performance compared to LXMERT. However, TPS of CFE is much better than the peer in LXMERT. Results demonstrate that CFE is enough for our method.

Table A4: Comparison between the peer in LXMERT. The input resolution is $336 \times 336$ and the number of visual tokens fed into LLM is 144. $\sharp$ represents the multi-level language-guide visual projector.

| Method | TPS | VQA$^{v2}$ | GQA | VQA$^{T}$ | OCRB |
|---|---|---|---|---|---|
| LXMERT (Tan & Bansal, 2019) | 20.2 | **78.7** | 61.8 | 57.6 | **299** |
| LVP | **25.3** | 78.5 | **62.0** | **58.0** | 298 |
| LXMERT$^{\sharp}$ (Tan & Bansal, 2019) | 15.1 | 79.0 | **62.6** | 58.8 | **319** |
| LVP$^{\sharp}$ | **24.2** | **79.2** | 62.5 | **58.9** | 317 |

**Ablation study on $N_q$.** We ablate the influence of $N_q$, the number of visual toekns fed into LLM. As shown in the Table A5, we can see that when $N_q$ is less than 144, model performance improves as $N_q$ increases. However, when $N_q$ is larger than 144, the improvement is limited. $N_q = 256$ is better than $N_q = 324$ on VQAv2 and GQA. We attribute to that when $N_q$ is enough large, visual tokens fed into LLM are redundant.

Table A5: Ablation study on the visual tokens fed into LLM $N_q$.

| Method | VQA$^{v2}$ | GQA | VQA$^{T}$ | OCRB |
|---|---|---|---|---|
| 36 | 75.2 | 60.6 | 55.8 | 264 |
| 64 | 77.2 | 61.4 | 57.1 | 283 |
| 128 | 77.7 | 61.5 | 57.5 | 288 |
| 144 | 78.5 | 62.0 | 58.0 | 298 |
| 256 | 78.8 | 62.4 | 58.7 | 306 |
| 324 | 78.4 | 62.1 | 58.7 | 309 |

**Influence of SigLIP and Qwen2.5.** In this section, we ablate the effectiveness of SigLIP-ViT-L and Qwen2.5-7B. Results are shown in the Table A6. Both SigLIP and Qwen2.5-7B improve the model performance. It should be noted that Qwen 2.5-7B is more effective. Compared with Vicuna-7B, Qwen2.5-7B obtains 0.9% (79.4% vs. 78.5%), 1.1% (63.1% vs. 62.0%), 0.8% (58.8% vs. 58.0%), and 13 (311 vs. 298) improvement on four benchmarks under the normal input resolution settings, respectively. In the scenario of high-resolution, Qwen2.5-7B and SigLIP achieves the consistent improvement.

Table A6: Results of SigLIP (Zhai et al., 2023) and Qwen2.5 (Qwen Team, 2024). The normal resolution is $336 \times 336$ and high-resolution is $1088 \times 1088$. The $N_q$ of normal resolution and high resolution are 144 and 954, respectively.

| Vision Model | LLM | VQA$^{v2}$ | GQA | VQA$^{T}$ | OCRB |
|---|---|---|---|---|---|
| CLIP | Vicuna-7B | 78.5 | 62.0 | 58.0 | 298 |
| SigLIP | Vicuna-7B | 78.8 | 62.3 | 58.5 | 302 |
| CLIP | Qwen2.5-7B | 79.4 | 63.1 | 58.8 | 311 |
| SigLIP | Qwen2.5-7B | 79.5 | 63.5 | 59.2 | 319 |
| *High-resolution Setting* | | | | | |
| CLIP | Vicuna-7B | 81.0 | 64.2 | 68.0 | 484 |
| SigLIP | Vicuna-7B | 81.2 | 64.6 | 68.1 | 492 |
| CLIP | Qwen2.5-7B | 81.6 | 64.7 | 68.8 | 502 |
| SigLIP | Qwen2.5-7B | **81.8** | **64.9** | **69.0** | **511** |

### A.3 RESPENTATION OF VISUAL TOKENS

In this section, we discuss the representation of visual tokens fed into LLM. We still take the Vicuna-7B as the LLM. In order to facilitate the visualization, we set the input resolution to $112 \times 112$. The number of visual tokens fed into the LLM of the linear projector, resampler, LDPv2, and our LVP is 64, 16, 16, and 16, respectively. The visualization results are displayed in Figure A1. We can see that the concept of "wave" is allocated only one visual token (red box), causing the model to focus on the "surfer". However, from the attention map of the linear projector, we can find that "wave" should be allocated several visual tokens. As for our LVP, it can be observed that the proportion of the visual tokens representing "wave" is much higher than that of the resampler and LDPv2, effectively aligning the visual tokens and input text. The visualization results are in line with our motivation.

### A.4 QUALITATIVE RESULTS

In this section, we display the qualitative results of our LVP. Here, we adopt LLaVA1.5 with Vicuna-7B. We visual the output of TokenPacker (Li et al., 2024c) and LVP in Figure A2, including two tasks: VQA and OCR. It can be seen that the output of our LVP is more accurate than the output of TokenPacker, demonstrating the superiority of our LVP.

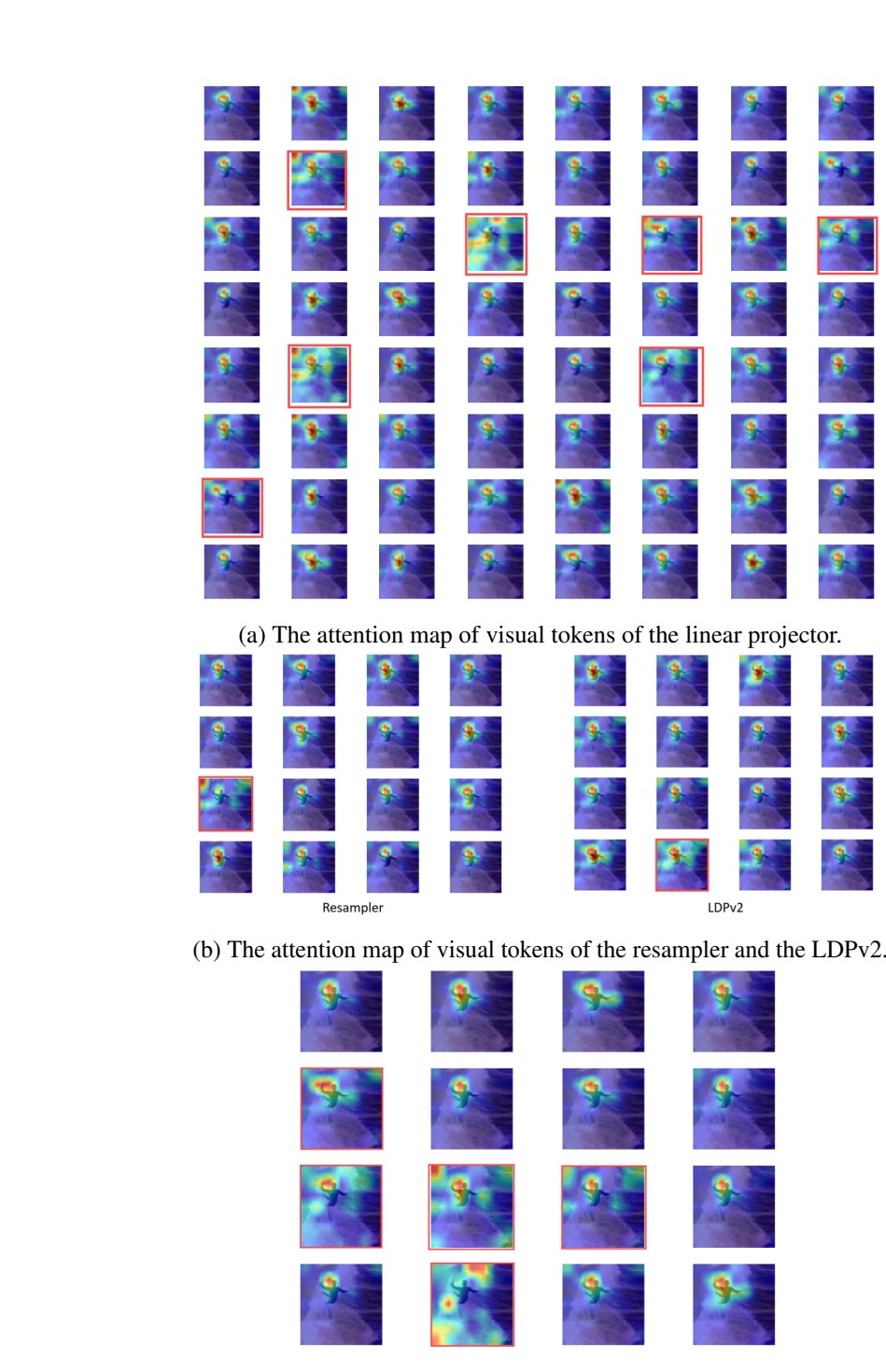

(a) The attention map of visual tokens of the linear projector.

(b) The attention map of visual tokens of the resampler and the LDPv2.

(c) The attention map of visual tokens of our LVP.

Figure A1: The attention map of visual tokens of different visual projectors. The input text is "What is happening with the surfer and the wave in the image?".

**LLaVA-LVP**

In this image, we see a group of people gathered around a dirt road, engaged in a lively game of frisbee. The person at the center of the action is a young boy who has just thrown the white frisbee towards another individual standing on the other side of the road. This person, dressed in **a blue shirt** and white skirt, appears to be waiting for the frisbee.The setting is a rural area with trees and buildings **with white roofs** visible in the background. The dirt road where they're playing suggests that it's a quiet, peaceful place away from the hustle and bustle of city life. Despite the simplicity of their surroundings, the joy and excitement of the game can be felt through the image.

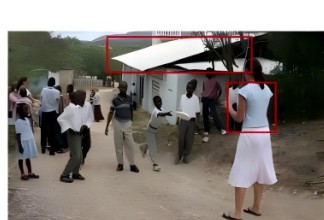

User：Describe the image in detail

**Tokenpacker**

In the heart of a rural setting, a group of children and adults gather on a dirt road. The children, dressed in vibrant clothing, are engaged in a lively game of frisbee. One child, in particular, is seen throwing the frisbee with enthusiasm.On the right side of the image, an adult woman stands out. She's casually dressed in **a white shirt** and white skirt, her attention focused on the children's game. Her stance suggests she might be supervising or simply enjoying the moment.The backdrop to this scene is a quaint white building with **a red roof**, adding a rustic charm to the overall setting. The building's location suggests it might be a community center or a local landmark.The image captures a moment of joy and camaraderie among the group, set against the backdrop of a simple yet beautiful rural landscape.

**LLaVA-LVP**

This image captures a scene in a kitchen, dominated by a white sink nestled into a wooden cabinet. The sink, pristine and clean, contrasts sharply with its surroundings. Above it, a faucet stands ready for use, while below it, a garbage disposal system waits to do its duty.To the left of the sink, a dishwasher stands closed, its surface gleaming under the light. On the right, a cabinet door hangs open, revealing a glimpse of the chaos within - **a jumble of dishes and bottles**, hinting at a busy day in the kitchen.**Scattered across the floor beneath the sink, towels lie in disarray**, adding to the sense of disorder. Above the sink, a window punctuates the wall, offering a view of the world outside. The overall scene paints a picture of a kitchen in need of some tidying up after a busy day of cooking and cleaning.

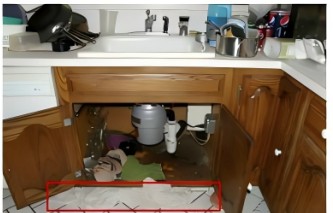

User：Describe the image in detail

**Tokenpacker**

The image captures a scene of a kitchen in the midst of a remodel. The central focus is a white sink, which is nestled within a wooden countertop. The countertop, with its light brown hue, contrasts with the white sink and backsplash.Beneath the sink, a metal pipe runs horizontally across the floor, serving as a conduit for the plumbing. This pipe is not just functional but also hosts an array of items that add character to the scene. A green rag lies atop the pipe, perhaps used in cleaning or cooking. Nearby, **a white towel hangs neatly**, ready for use.The floor beneath these fixtures is tiled, **providing a clean and bright backdrop to the scene**. On the right side of the image, a can of soda stands upright, its vibrant color adding a pop of contrast to the otherwise neutral tones of the room.

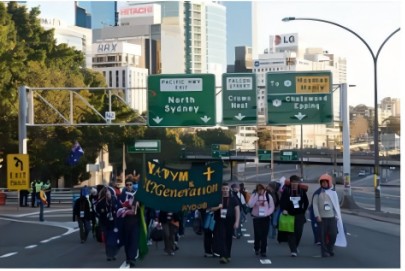

User：Where is the Falcon Street exit going to ?

**LLaVA-LVP**

**Crows Nest**

**Tokenpacker**

**North Sydney**

Ground Truth: **Crows Nest**

Figure A2: Visual comparison between TokenPacker and our LVP. We use red color to represent the accurate output and blue to denote the false output.

