# OpenReview forum: "LVP: Language-guide Visual Projector for Efficient Multimodal LLM"
_ICLR.cc/2025/Conference — Submitted to ICLR 2025_

### Official Review · Reviewer_hBbQ · 2024-10-31

**Soundness:** 3
**Presentation:** 2
**Contribution:** 2
**Rating:** 5
**Confidence:** 5

**Summary:**

This paper introduces a visual projector called LVP, designed to compress visual tokens for efficient multimodal
large language models (LLMs).  LVP is a language-guided visual projector in which text features serve as a guide for
selecting important visual tokens based on feature similarity. Extensive experiments are conducted to validate the performance of LVP.

**Strengths:**

1. The text-guided selection for visual token compression presents a novel approach to the task.
2. Compared to previous token compression methods, LVP achieves state-of-the-art performance
under identical experimental conditions.

**Weaknesses:**

1. The overall discourse of this paper closely resembles the previous research on TokenPacker,
particularly in the second and third paragraphs of the Introduction, the first and second paragraphs
of the Related Work section, and the descriptions of the datasets and main comparison results in the Experiments section.
This version of the paper requires substantial revisions to eliminate textual repetition.

2. In Figure 2, the visual attention map should indicate which layer in the LLM it corresponds to.

3. In the high-resolution experiments, this paper adopts the same dynamic image slicing method proposed in TokenPacker-HD with resolutions of 1088x1088 and 1344x1344. If this is the case, it should be clarified.

**Questions:**

I noticed that in Table 3, the performance of 64 tokens (9x compression ratio) utilizing multi-level features achieves comparable results to that of the MLP with 576 tokens, which is impressive.

I am concerned whether the code will be made publicly available to facilitate better reproduction of results.
I encourage the authors to release the code to benefit the field.

#####Post-rebuttal#####

After reviewing the feedback and comments from other reviewers, I have decided to lower my scores for the following reasons:

1. The presentation of this paper largely borrows from the previous paper, TokenPacker, which the authors have acknowledged. Although revisions have been made, many details still exists. It is unacceptable for an academic paper. Additionally, the overall presentation is rough, particularly for figures. I think  this paper is  not adequately prepared for publication.

2. The novelty of this work is limited, despite achieving better performance. The core contribution is  a combination of language-guided methods derived from Grounding DINO  and TokenPacker.  This issue has also been pointed out by Reviewer Note R.

---

> ### Author Response · Authors · 2024-11-22
> **Response to Reviewer hBbQ**
>
> **Q1:The overall discourse of this paper closely resembles the previous research on TokenPacker, particularly in the second and third paragraphs of the Introduction, the first and second paragraphs of the Related Work section, and the descriptions of the datasets and main comparison results in the Experiments section. This version of the paper requires substantial revisions to eliminate textual repetition.**
>
> REPLIES: We fully agree with the reviewer and have thoroughly revised the paper to eliminate textual repetition. The specific modifications include  1) the second and third paragraphs of the Introduction, the Related Work section, the dataset section, and the normal resolution comparison in the Experiments section.
>
> **In Figure 2, the visual attention map should indicate which layer in the LLM it corresponds to.**
>
> REPLIES: We would like to further explain this concern.  We adopt an approach similar to R-GAE in DeCo to get the attention map.
> Specifically, we first construct a Text-to-Visual map $M_t \in R^{N_I \times N_q}$, where $N_I$ denotes the number of visual tokens outputted by the visual encoder and $N_q$ stands for the number of visual tokens fed into LLM.  $M_t$ is initialized to an identity matrix. For each layer in the projector, an attention map is obtained by utilizing the gradients to average across the attention heads for the resampler and LVP. For Linear projector and LDPv2, an attention map is obtained by adopting the gradients of each layer.
> For generation time step $t$, we can propagate the $M_t$ from the projector's first layer to its last layer. Finally, we average the step $t$ and average the $M_t$ to get the final attention map. **We have added this clarification in Appendix A.1**.
>
> **Q3:In the high-resolution experiments, this paper adopts the same dynamic image slicing method proposed in TokenPacker-HD with resolutions of 1088x1088 and 1344x1344. If this is the case, it should be clarified.**
>
> REPLIES: We do not use the dynamic image-slicing method proposed in TokenPacker-HD. Our high-resolution image procession is the same as that in LLaVA1.5-HD.
>
> **R4Q4: I am concerned whether the code will be made publicly available to facilitate better reproduction of results. I encourage the authors to release the code to benefit the field.**
>
> REPLIES: We totally agree with the reviewer's comments. We have provided the code in the supplementary material. After the anonymous review, we plan to open-source the code to GitHub.

---

> > ### Author Response · Authors · 2024-11-25
> > **Looking forward to the feedback on our responses**
> >
> > Dear Reviewer hBbQ
> >
> > We would like to express our sincere thanks again for the time and effort you spent on our paper. Since the discussion deadline is approaching, we kindly look forward to your feedback on our responses. We would be happy to discuss this further if there are still some open concerns.
> >
> > Best regards
> >
> > Authors

---

> > ### Comment · Reviewer_hBbQ · 2024-11-25
> >
> > Thank you to the authors for their efforts in addressing my concerns; however, some issues remain:
> >
> > 1. **Response to R.Q1**:  Although the authors have revised the paper, several aspects still bear a strong resemblance to TokenPacker paper.  In Table 2, the annotations †，♯，and  §  are directly  appear to be same to those in the TokenPacker paper but lack the explanations. Additionally, the presentation in the conclusion section is also notably similar to that found in TokenPacker. I believe this raises concerns regarding the academic rigor of the paper. **Reviewer oteR also highlighted this issue.**
> >
> > 2. **Response to R.Q3**: In your response, you claim that the high-resolution image processing is the same as that in LLaVA1.5-HD. However, the number of tokens of the method corresponds to TokenPacker as shown in Table 2, which I believe is incorrect. Different slicing methods yield different numbers of slicing patches, leading to discrepancies in the number of tokens from a statistical perspective. I think this requires further clarification.
> >
> > 3. **Questions on  the results of LLaVA-OV**: I have noted the response to Reviewer GoFW, in which the authors provided results based on LLaVA-OV. To the best of my knowledge, the LLaVA-OV dataset has not been fully released, particularly regarding the multi-image and video components.  Would you like to share how did you reproduce the results using your method.

---

> ### Author Response · Authors · 2024-11-26
> **Further discussion with Reviewer hBbQ**
>
> **Q1:Concerns about the presentation**
>
> REPLIES: Thank you for your reply. In the latest revision, we have carefully revised the full text, including the abstract, main results, and conclusion, to eliminate textual repetition. In addition, we have corrected the symbols in Table 2 and added descriptions of them in Table 2. We tried our best to improve our presentation and double-checked the paper to ensure academic rigor. We believe the latest version is different from that of TokenPacker.  If you still think there exists the presentation similarity, **please point them out**. We will try our best to revise them.
>
> **Q2:Questions about dynamic image slicing in TokenPacker**
>
> REPLIES: Thanks for your reply, here we provide further clarification on the number of tokens of the method. The number of visual tokens in our method is determined manually, i.e., the first $N_q$ visual tokens are input to the LLM. As we mentioned in the paper (line 190-line 191), the visual tokens of our method can be flexibly adjusted. The number of visual tokens input into LLM after processing of high-resolution images in LLaVA1.5-HD is much higher than $N_q$. In order to be able to make a fair comparison with the TokenPacker, we set $N_q$ equal to the number of visual tokens in the TokenPacker.
>
> **Q3:Question about the LLaVA-OV**
>
> REPLIES: Here we provide further clarification on the reproduction of LLaVA-OV results. In the latest LLaVA-OneVision Github, the authors added the construction of LLaVA-OneVision data, including single-image, multi-image, and video data. We followed their tutorial to prepare the data and perform the corresponding experiments. The authors have supplemented the preparation details in https://github.com/LLaVA-VL/LLaVA-NeXT/blob/main/scripts/train/README.md. Hope the above replies can solve your concerns.
>
> **Question about rating**
>
> We notice that the reviewer lowers the rating and want to further discuss with reviewers. The questions about the novelty mentioned by Reviewer oteR are inappropriate, which we have fully discussed in our initial submission and we have given detailed responses to this. However, we think our method can bring insights to the community as mentioned by Reviewer goFW and sWSo. Regarding your concerns, we have tried our best to solve them. We hope you can reconsider our paper. Thanks again!

---

> ### Author Response · Authors · 2024-11-26
> **Response to the reasons for lowing rate**
>
> We would like to further discuss with reviewers the reasons for lowing rate.
>
> **Q1:Concerns about the presentation**
>
> REPLIES: As answered above, we have fully advised the paper and hope the revision can help resolve the concerns. Besides, we don't acknowledge that the presentation of this paper largely borrows from the previous paper, TokenPacker. We revised these sections to eliminate the text similarity mentioned by the Reviewer hBbQ.
>
> **Q2:Concerns about the novelty**
>
> REPLIES: As we discussed with Reviewer oteR. We would like to discuss this with the reviewer.
>
> **Motivation and method**. In the motivation of our method, we found that the visual token outputted from the visual projector is not aligned with the textual token, **which has not been discussed in previous work**. Based on this finding, the direct way is using the textual features to select the visual tokens since the number of visual tokens is much larger than textual tokens. We depend on this and make the idea work and hope that this idea can be an effective trick for future design of visual projectors.  The motivation and methods are acknowledged by Reviewer goFW and sWSo.
>
> **Difference**. In order to highlight the difference with the existing methods, we have compared Grounding-DINO with our method in the initial submission. Our method is a plug-and-play module but the token selection in Grounding-DINO highly depends on its whole architecture. Besides, the selection of visual tokens in Grounding-DINO is a smaller part of the contribution, and its overall performance relies on the heavy decoder and the visual encoder. We also conducted experiments to demonstrate that such a heavy architecture does not achieve better performance.
>
> For the differences between TokenPacker and our LVP, we have elaborated above, i.e., textual features to select important visual tokens as well as DeformableAtt.
>
> Therefore, our method **is not a combination of language-guided methods derived from Grounding DINO and TokenPacker** but an effective visual projector, aligning text tokens and visual tokens.

---

> ### Comment · Reviewer_hBbQ · 2024-11-26
> **Reasons for reducing rating score.**
>
> Initially, I think this paper is impressive and assigned a preliminary score of 6 due to its commendable performance as a visual projector in MLLM for token compression.
> **After reviewing the comments from other reviewers and carefully check the paper and code, I changed my assessment.** **I identified several significant issues**, which I summarize in four aspects:
>
> **1.** **About the Novelty**.
>
> I still believe that the proposed method is a combination of existing techniques. There are clear evidences:
>
> (a) The language-guided visual token selection outlined in Equ. (2) is identical to the language-guided query selection in Algorithm 1 of Grounding DINO.  and **the code is also largely the same**.  Specifically, the implementation of the **class ContrastiveEmbed** in LVP is nearly identical to the same class ContrastiveEmbed in the Grounding DINO code.
>
> (b) Furthermore, the text-to-image&image-to-text attention mechanisms used to enhance cross-modal features are also derived from Grounding DINO.
>
> (c) The DeformAtt is subquently adopted for the enhanced query, while previous methods utilized the original attention mechanism. Additionally, the use of multi-level features is similar to those in TokenPacker, which employs same levels of 12, 16, 22, and 23.
>
>
> **2.** I believe that  **the token numbers**  **(954/1393/619/347)** **presented in Table 2 for high-resolution methods are incorrect.**
>
> I checked the code and found that the high-resolution settings default to the AnyRes settings in LLaVA-1.5-HD, as acknowledged by the authors.  However, the differing patch slicing schemes lead to varying patch counts for each image, resulting in different token numbers.  The authors claim that the number of tokens can be set to any arbitrary value, which seems implausible unless it changes dynamically for each image with a calculating value. Nevertheless, it is difficult to set a specific token number (954/1393/619/347) with different dynamic slicing.
>
> Furthermore, I found the value is fixed (e.g., 144/64) in the code. Hence, I believe this assertion by the authors is incorrect.
>
> Additionally, the authors did not specify that they adopted the partitioning strategy from LLaVA-HD at all.
>
> **3.** The paper **lacks rigor** and **exhibits numerous areas of ambiguity**, although the authors have revised the paper many times:
>
> (a) 	In the multi-level language-guided visual projector, the authors mention "$N_a$ x 4" but do not clarify how to set $N_a$ and its relationship with original $N_q$ in Equ.(2).  The relevant code for the multi-level implementation is also not provided.
>
> (b)  The authors claimed in their response to Reviewer goFW that they utilized the first two layers of BERT, initialized with BERT's weights as a text encoder; however, this detail is not mentioned in the paper at all.
> …
>
> **4.** **About Presentation Quality**:  The figures in the paper appear rough and are not adequately prepared for publication. I think that the submission has been rushed, resulting in a lack of overall quality.
>
>
> **In conclusion**, considering these factors, I firmly believe that this paper is not suitable for publication, below the acceptance.

---

> ### Author Response · Authors · 2024-11-27
> **Further discussion with Reviewer hBbQ[1/3]**
>
> Thank you for your detailed response. We believe that the discussion is meaningful and provides valuable suggestions to further improve the quality of the paper. We would like to provide further explanation on a few points raised in the comments raised by the reviewer.
>
> Before we start the discussion, we think **we have reached a consensus** on some open concerns: 1) **Presentation similarity**. After several rounds of discussion with the reviewers, we have resolved the presentation similarity issue, which can be reflected in the latest version of the revision. 2) **The LLaVA-OV data issue has also been resolved**. We are very happy that the above issues have been resolved through our efforts. Let's go further and discuss the current concerns: 1) novelty. 2) the number of visual tokens for high-resolution.
>
> For **novelty**, we would like to elaborate on two aspects.
>
> 1) **Motivation**. The motivation of our paper is to align the visual tokens and textual tokens since the misalignment problem leads to the loss of important cues in the text (instruction). This is clearly demonstrated in Figure 2 of our paper. **This problem is first raised in our work for the design of an effective visual projector**, which we would like to agree with the reviewers (Reviewer sWSo and goFW have acknowledged this point), **as it is crucial to explore the implications of subsequent approaches**. We hope the community can give a certain level of concern to the finding in our motivation. Solving this problem has also been shown to be useful in our experiments (see Table 5) for improving the performance of the efficient visual projector, even beyond MLP (**our approach is the first among the existing visual projectors**).
>
> 2) **Method**. Based on this Motivation, we need to find a method that can align visual tokens and textual tokens. **We believe that a good method should not need a fancy structure, it should be simple and elegant**. The number of visual tokens and the redundancy of features are significantly higher than textual tokens, so the most intuitive way is to use textual tokens as a guide to select the visual tokens. We designed LVP based on this.
>
> Through the above narrative, it can be seen that the innovation and the significance of our method consist of two aspects: 1) **We found a factor that hinders the visual projector**. 2) In order to solve this problem, we propose our method. **We think it is useful (also acknowledged by Reviewer sWSo and goFW) and can bring some insight to the community**. This is something we hope to discuss with the reviewers to exchange ideas.
>
> Let's go back to the difference between Grounding-DINO and our approach, which is what we consider an important point of contention. We would like to elaborate on the following aspects.
>
> 1) As we have elaborated above, our method focuses on the need to solve the problem of text and token misalignment that we have identified. To solve this problem, we propose to use textual features to select visual features, while token selection in Grounding-DINO is a kind of training trick, which is inconsistent with the problem we are trying to solve. **For MLLM, our approach is the first one, which is also recognized by reviewer sWSo**. Also, **the strength in your review also acknowledges this point that our approach is novel to this task**.
>
> 2) The formulas you mentioned in Eq.2 are consistent with those in grounding-dino, which we need to clarify: 1) We normalize the features and then compute the similarity, while Grounding-Dino does not. 2) In the code, we have added learnable bias, which is a training trick, but Grounding-DINIO does not.  In addition, **we believe that the code implementation should not be part of the discussion of innovation** since different engineers can have different implementations for the same task. **Another important point: we discussed the differences between our approach and Grounding-DINO's approach in the initial submission**, and conducted the experiments to validate that Grounding-DINO's architecture is not optimal for our approach. Our method is very lightweight, which is in line with the expectations of a visual projector. The above clarification is enough to prove that we have discussed this issue thoroughly and verified it experimentally.
>
> 3) Image-to-text attention and Text-to-Image attention, as mentioned by the reviewer sWSo,  **is a step that is adopted in many methods because it is helpful for cross-modal feature learning**, but as we demonstrated in our experiments, the existing cross-modal methods are too heavy, and the lightweight structure of our method is sufficient for our method. In addition, we also clarify this point in the initial submission and have given a full discussion on it.

---

> ### Author Response · Authors · 2024-11-27
> **Further discussion with Reviewer hBbQ[2/3]**
>
> 4) This point about DeformAtt needs further discussion. First of all, **the step of selecting visual tokens is not available in TokenPacker and Mini-Gemini**, in other words, they reduce visual tokens by downsampling, which is fundamentally different from our approach. In addition, **the visual tokens after text selection cannot be directly inputted into LLM, which will lead to the loss of visual information**, so it is necessary to design a module that can aggregate the visual features, and the attention module is undoubtedly the most suitable. We still believe that designing fancy structures and finding a suitable design justification for this will not increase the impact of the method, so we compare the difference between attention and DeformAtt. DeformAtt is more suitable. We have fully compared the difference between our approach and TokenPacker and Mini-Gemini in terms of the use of attention in the introduction and related work **in the initial submission**: the previous two are aggregating local features, while our approach is aggregating global features.
> 5) For the choice of different layers in the use of multi-level features, we have already explained it explicitly in the paper. If it is not mentioned, we think it is appropriate as a negative justification.
>
> As described in the reviewer guide of ICLR, **"Be mindful of potential biases and try to be open-minded about the value and interest a paper can hold for the entire ICLR community, even if it may not be very interesting for you"**. We hope the reviewer can consider our paper for its significance to the ICLR community, **including the motivation, findings proposed in our paper, our method, and the experimental results**. We appreciate the efforts and time the reviewer spent in the discussion again!

---

> ### Author Response · Authors · 2024-11-27
> **Further discussion with Reviewer hBbQ[3/3]**
>
> For the concern about the **token numbers presented in Table 2 for high-resolution methods**, we further elaborate on this in more detail and apologize for not solving your confusion in the previous discussion.
>
> Given a high-resolution image, LLaVA-HD first splits the image into different patches and each patch is fed into the visual encoder. The visual encoder outputs a sequence of visual tokens. We use ${P}_{i=1}^N$ to represent the sequence of visual tokens and $N$ is the number of patches. Besides, LLaVA-HD resizes the original high-resolution image to the size the visual encoder can process. Here we use $P_H$ the denote the the visual token of the high-resolution image.
>
> Finally, LLaVA-HD concatenate the ${P}_{i=1}^N$ and  $P_H$. We use $P_C$ to stand for the concatenated visual tokens. $P_C$ is the visual input of our LVP. We use the text feature as a guide to select the Top $N_q$ ($N_q$ is much smaller than the number of $P_C$) visual tokens from $P_C$ based on the similarity between visual features and text features. We set $N_q$ to 954/1393/619/347, respectively. Hope this clarification can resolve your concerns.
>
> **We have supplemented this implementation to the appendix, including the specification of LLaVA-HD**. Thanks for pointing it out.
>
> For the concern about the **presentation**.
>
> 1) We further explain it. First, the definition of $N_a$ is explicitly explained in the paper. See **line 267: we select the top $N_a$ visual tokens as Eq. 2.**, indicating that we follow Eq. 2 to select the visual tokens and output $N_a$ visual tokens. Simple understanding: $N_a$ is equal to $N_q$. However, we use different notions to better describe our method. In our paper, $N_q$ is used to denote the number of visual tokens generated by a single layer but $N_a$ represents the number of visual tokens generated by a single layer in the multi-layer LVP.
>
> 2) For the implementation in BERT, **we have added it to our paper (see implementation detail)**.
>
> 3) Reviewer claims that **"I think that the submission has been rushed, resulting in a lack of overall quality."** is subjective. **We indeed spend a lot of time doing this work and writing the paper. We follow the reviewers' comments to make our paper move in a good direction.** Both Reviewer sWSo and goFW acknowledge the presentation of our paper. We will follow the reviewer's comment to polish our figure.
>
> 4) For other ambiguities, we will try our best to revise them.
>
> We hope that our responses can resolve the reviewer's concerns and reconsider our paper based on the feedback to the comments. We also look forward to your insightful comments to make our paper better and really appreciate your time and efforts.

---

> > ### Author Response · Authors · 2024-12-02
> > **Looking forward to the feedback**
> >
> > Dear Reviewer hBbQ
> >
> > We would like to express our sincere thanks for the time and effort you spent on our paper. Since the discussion deadline is approaching, we kindly look forward to your feedback on our latest responses.
> >
> > Best regards
> >
> > Authors

---

### Official Review · Reviewer_goFW · 2024-11-03

**Soundness:** 4
**Presentation:** 4
**Contribution:** 4
**Rating:** 6
**Confidence:** 5

**Summary:**

Emergency review for Submission #6390

This paper presents the Language-guide Visual Projector (LVP) for multimodal Large Language Models (MLLMs). Existing visual projectors in MLLMs have issues like heavy computational burden or feature misalignment. LVP uses text features to guide the selection of important visual tokens, with a lightweight text encoder, a cross-modal feature enhancement module, and a deformable attention module. In addition, the authors propose a multi-level language-guide visual projector to generate the visual tokens from different stages of the encoder. Experiments show that LVP compresses visual tokens by 75% - 95% and achieves good performance across various MLLM benchmarks.

**Strengths:**

The advantages of the submission are obvious:

S1. The research on reducing tokens is quite useful. The methodology of the submission is also technically sound.

S2. The performance of the proposed is very good. It beats a lot of baselines. The MME score is also very high.

S3. This paper is also well-written and is easy-to-follow.

S4. According to the visualization, the performance of author's method is also sound.

**Weaknesses:**

There are three weaknesses from my point of view.

W1. Compared with image tasks, the research on video tasks can better highlight the authors' motivation. We know, there is a lot of redundant information in video. Therefore, I suggest the authors extend their paper to video tasks as in VideoLLaMA and LLaVA-OV.

W2. I suggest the authors present TPS in Tab.1 and Tab.2.

W3. As a submission to ICLR, the paper should have some discussions on the representations of the visual token. Some related experiments need to be conducted.

**Questions:**

I have some questions.

Q1. Why the datasets evaluated in Tab.1 are not in line with those of Tab.2? Can you report them in total?

Q2. What is the influence of your method applied to video tasks?

Q3. The idea of using text features has already been discussed in InstructBLIP. What are the advantages of your technique compared with InstructBLIP?

Q4. How $N_q$ influence the last model performance?

Q5. Why deformable attention is utilized. How about a regular self-attention?

Q6. Is it possible to quantitatively or qualitatively analyze the representation of tokens input to LLM?

Q7. The visual feature encoder and LLM used by the author are relatively outdated. I know this is for a fair comparison. I want to know how good the performance would be if you consider SigLIP and QWEN2.5 and develop a version under the high-resolution setting.

Justification on rating:

I think the proposed method is useful. I have no firm reasons to reject this paper. I will stick to a positive one if all my concerns are addressed.

---

> ### Author Response · Authors · 2024-11-22
> **Response to Reviewer goFW [1/2]**
>
> **W1:Compared with image tasks, the research on video tasks can better highlight the authors' motivation. We know, there is a lot of redundant information in video. Therefore, I suggest the authors extend their paper to video tasks as in VideoLLaMA and LLaVA-OV.**
>
> REPLIES: We appreciate the reviewer’s expertise and insight. Following this constructive comment, we take the LLaVA-OneVision (OV) as the baseline. We replace the projector in LLaVA-OV with our LVP. The training data is the same as that in LLaVA-OV. We report a score out of 5 for VideoDetailCaption (VideoDC), and VideoChatGPT while other results are reported in accuracy. All
> results are reported as 0-shot accuracy. The results are displayed below. LLaVA-N-Video is LLaVA-Next-Video and IXC-2.5 denotes InternLM-XC-2.5.
>
> |Model|ActNet-QA|EgoSchema|MLVU|MVBench|NextQA|PercepTest|SeedBench|VideoChatGPT|VideoDC|VideoMME|LongVideoBench|
> |----|----|----|----|----|----|----|----|----|----|----|----|
> |VILA-40B|58.0|58.0|-|-|67.9|54.0|-|3.36|3.37|60.1/61.1|-|
> |PLLaVA-34B|$\bf 60.9$|-|-|58.1|-|-|-|3.48|-|-|-|
> |LLaVA-N-Video-34B|58.8|49.3|-|-|70.2|51.6|-|3.34|3.48|52.0/54.9|50.5|
> |IXC-2.5-7B|52.8|-|37.3|$\bf 69.1$|71.0|34.4|-|3.46|3.73|55.8/58.8|-|
> |LLaVA-N-Video-32B|54.3|60.9|65.6|-|77.3|$\bf 59.4$|-|3.59|3.84|$\bf 60.2$/$\bf 63.0$|-|
> |LLaVA-OV-0.5B|50.5|26.8|50.3|45.5|57.2|49.2|44.2|3.12|3.55|44.0/43.5|45.8|
> |LLaVA-OV-LVP-0.5B|51.0|28.0|51.0|46.3|57.9|50.3|44.9|3.55|3.77|45.9/44.7|47.0|
> |LLaVA-OV-7B|56.6|60.1|64.7|56.7|79.4|57.1|56.9|3.51|3.75|58.2/61.5|56.4|
> |LLaVA-OV-LVP-7B|57.3|$\bf 61.0$|$\bf 65.8$|57.8|$\bf 80.3$|58.3|$\bf 57.6$|$\bf 3.70$|$\bf 3.88$|59.9/$\bf 63.0$|$\bf 57.8$|
>
> The number of visual tokens fed into LLM in LLaVA-OV is Z$\times$196, where Z is the sampled frame per video. The number of visual tokens fed into LLM in LLaVA-OV-LVP is Z$\times$98.
>
> LVP improves the model performance on all 11 benchmarks, showing its advantages in the video tasks. LVP gains 1.2\% (47.0\% vs. 45.8\%) and 1.4\% (57.8\% vs. 56.4\%) for 0.5B and 7B model on LongVideoBench,
> demonstrating its strength in long video understanding. Besides, LLaVA-OV-LVP-7B achieves better LLaVA-N-Video-32B on ActNet-QA, EgoSchema, MLVU, NextQA, VideoChatGPT, VideoDC, and LoneVideoBench, indicating that our LVP is an effective visual projector for video tasks. **We have added the video experiment to Section 4,3 (Video Benchmarks).**
>
> **W2: I suggest the authors present TPS in Tab.1 and Tab.2.**
>
> REPLIES: As the reviewer will note, we have presented TPS in Tab.1 and Tab.2.
>
> **W3:As a submission to ICLR, the paper should have some discussions on the representations of the visual token. Some related experiments need to be conducted.**
>
> REPLIES: Following this constructive comment, we have supplemented the discussion on the representations of the visual token in Appendix A.3.
>
> **Q1:Why the datasets evaluated in Tab.1 are not in line with those of Tab.2? Can you report them in total?**
>
> REPLIES: We totally agree with the reviewer. We report the evaluation datasets in both Tab.1 and Tab.2.
>
> **Q2:What is the influence of your method applied to video tasks?**
>
> REPLIES: Answered with the W1 above.
>
> **Q3:The idea of using text features has already been discussed in InstructBLIP. What are the advantages of your technique compared with InstructBLIP?**
>
> REPLIES: The advantages of our method lie in two ways: 1) Quality of visual tokens. As described in DeCo [1], Q-Former in InstructBLIP is difficult to learn well as a semantic extractor, which leads to poor feature extraction of visual tokens as well as loss of visual features. The complement of task-relevant knowledge provided by the textual features cannot compensate for the shortcomings of the visual features. Our approach uses textual features as guidance to select important visual tokens instead of double feature extraction in InstructBLIP, so the visual tokens have better features. We avoid feature loss by supplementing key visual features with deformable attention. 2) Our approach uses the Cross-modal feature enhancement module to enhance the cross-modal feature learning, which is more conducive to multi-modal feature interaction than using a single Self-Attention in InstructBLIP.
>
> [1] Yao L, Li L, Ren S, et al. DeCo: Decoupling Token Compression from Semantic Abstraction in Multimodal Large Language Models[J]. arXiv preprint arXiv:2405.20985, 2024.

---

> ### Author Response · Authors · 2024-11-22
> **Response to Reviewer goFW [2/2]**
>
> **Q4: How $N_q$ influence the last model performance?**
>
> REPLIES: We conduct the ablation study to evaluate how $N_q$ influences the last model performance. As shown in the Table below, we can see that when $N_q$ is less than 144, model performance improves as $N_q$ increases. However, when $N_q$ is larger than 144, the improvement is limited. $N_q$=256 is better than $N_q$=324 on VQA$^\text{v2}$ and GQA. We attribute to that When $N_q$ is enough large, visual tokens fed into LLM are redundant. **We have supplemented the results in Appendix A.2 (Line 867-875)**.
>
> |$N_q$|VQA$^\text{v2}$ |GQA |VQA$^\text{T}$ |OCRB |
> |----|----|----|----|----|
> |36|75.2|60.6|55.8|264|
> |64|77.2|61.4|57.1|283|
> |128|77.7|61.5|57.5|288|
> |144|78.5|62.0|58.0|298|
> |256|78.8|62.4|58.7|306|
> |324|78.4|62.1|58.7|309|
>
> **Q5:Why deformable attention is utilized? How about a regular self-attention?**
>
> REPLIES: The reason for utilizing deformable attention is as follows: Direct use of the selected visual token causes loss of image features, so we introduce deformable attention to fuse the key features into the selected visual token.
>
> We conduct the experiment to compare deformable attention and the regular attention. The results are shown below.
>
> |Attention module|TPS|VQA$^\text{v2}$ |GQA |VQA$^\text{T}$ |OCRB |
> |----|----|----|----|----|----|
> |None|26.0|75.4|59.5|55.7|269|
> |Regular attention|25.0|78.0|61.7|57.3|291|
> |Deformable attention|25.3|78.5|62.0|58.0|298|
>
> It can be observed that without the attention module, the performance declines, indicating integrating the key features into the selected visual tokens is necessary. Compared with regular attention, deformable attention achieves better performance. We attribute the reason to the fact that regular attention integrates the redundant visual features from the visual encoder into the selected visual tokens, while deformable attention adopts the learnable keys to fuse the important visual features into the selected visual tokens, improving the feature learning efficiency.
>
> **Q6:Is it possible to quantitatively or qualitatively analyze the representation of tokens input to LLM?**
>
> REPLIES: Answered with the W3 above.
>
> **Q7:The visual feature encoder and LLM used by the author are relatively outdated. I know this is for a fair comparison. I want to know how good the performance would be if you consider SigLIP and QWEN2.5 and develop a version under the high-resolution setting.**
>
> REPLIES: We appreciate the reviewer’s expertise and insight. We ablate SigLIP-ViT-L and Qwen2.5-7B. The results are shown in the Table below. Both SigLIP and Qwen2.5-7B improve the model performance. It should be noted that Qwen 2.5-7B is more effective. Compared with Vicuna-7B, Qwen2.5-7B obtains 0.9% (79.4% vs. 78.5%), 1.1% (63.1% vs. 62.0%), 0.8% (58.8% vs. 58.0%), and 13 (311 vs. 298) improvement on four benchmarks under the normal input resolution settings, respectively. In the scenario of high-resolution, Qwen2.5-7B and SigLIP achieve consistent improvement.
>
> |Vision Model|LLM|VQA$^\text{v2}$ |GQA |VQA$^\text{T}$ |OCRB |
> |----|----|----|----|----|----|
> |CLIP|Vicuna-7B|78.5|62.0|58.0|298|
> |SigLIP|Vicuna-7B|78.8|62.3|58.5|302|
> |CLIP|Qwen2.5-7B|79.4|63.1|58.8|311|
> |SigLIP|Qwen2.5-7B|$\bf79.5$|$\bf63.5$|$\bf59.2$|$\bf319$|
>
> The results of the ablation study under the high-resolution setting are shown in the Table below.
>
> |Vision Model|LLM|VQA$^\text{v2}$ |GQA |VQA$^\text{T}$ |OCRB |
> |----|----|----|----|----|----|
> |CLIP|Vicuna-7B|81.0|64.2|68.0|484|
> |SigLIP|Vicuna-7B|81.2|64.6|68.1|492|
> |CLIP|Qwen2.5-7B|81.6|64.7|68.8|502|
> |SigLIP|Qwen2.5-7B|$\bf81.8$|$\bf64.9$|$\bf69.0 $|$\bf511$|
>
> **We have added the above results in Appendix.A.2 and supplemented a high-resolution version in the Table.2**.

---

> > ### Comment · Reviewer_goFW · 2024-11-24
> >
> > Thank your rebuttal. This method achieves good performance on both image and video tasks. I vote for accept.

---

> > > ### Author Response · Authors · 2024-11-24
> > > **Many thanks to Reviewer goFW**
> > >
> > > We would like to express our sincere thanks to the reviewer for the insightful feedback and support! Your suggestions are beneficial in improving the quality of our paper.

---

> > > > ### Comment · Reviewer_goFW · 2024-11-25
> > > >
> > > > I have found some typos. Remember to fix them.
> > > >
> > > > Table A5: Missing .
> > > >
> > > > Line818: Missing reference
> > > >
> > > > Numbers in Tab. 7 should be textbf
> > > >
> > > > I have also checked your code. It seems that the tokenizer used in LVP (BERT) differs from the LLMs (QWEN and LLaMA). Will it introduce misalignment?

---

> > > > > ### Author Response · Authors · 2024-11-26
> > > > > **Further discussion**
> > > > >
> > > > > **Q1: Writing typos**
> > > > >
> > > > > REPLIES: Thanks for pointing them out. We have fixed these typos in the latest revision.
> > > > >
> > > > > **Q2: Misalignment caused by different tokenizers**
> > > > >
> > > > > REPLIES: In our implementation, we initialize the weights of the text encoder using the first 2 layers in Bert. In order to avoid the conflicts caused by different tokenizers in different language models, we use the tokenizer of Bert in the LVP. It should be noted that the original tokenizer of LLM remains unchanged. In other words, we have two tokenizers in our method.
> > > > >
> > > > > Regarding your question, there is little or no effect between different tokenizers, since the LVP is just responsible for selecting important visual tokens. Besides, the training of LLaVA consists of two stages: 1) the pretraining of visual projector. 2) visual instruction tuning. Therefore, the features outputted by the text encoder in LVP can align with the knowledge of the whole model. Sure, If time allows, we will conduct the experiments to verify this point.

---

### Official Review · Reviewer_oteR · 2024-11-03

**Soundness:** 3
**Presentation:** 2
**Contribution:** 3
**Rating:** 5
**Confidence:** 5

**Summary:**

This work introduces a language-guided visual projector for multimodal large language models (MLLMs).  LVP first proposes a lightweight text encoder to extract text features, followed by the introduction of a cross-modal feature enhancement module and a multi-level language-guidance feature selection module to generate compressed tokens. Experimental results demonstrate the effectiveness of the proposed method.

**Strengths:**

1. The performance is impressive compared to existing methods.
2. The research focus is both interesting and effective.

**Weaknesses:**

1. The presentation of this paper is similar to that of TokenPacker.
2. In lines 245-251 on page 5, the feature injection using enriched features through cross-attention shares a similar idea with Mini-Genimi and TokenPacker, although you employ deformable attention in this work.
3. Minor issue: In Table 1, line 344, the value 53.9 should be in bold with the best performance, instead of 53.5.

**Questions:**

Considering the impressive performance, will the code be released publicly to facilitate reproduction of the results?

---

> ### Author Response · Authors · 2024-11-22
> **Response to Reviewer oteR**
>
> **Q1:The presentation of this paper is similar to that of TokenPacker.**
>
> REPLIES: We totally agree with the reviewer. We have revised the manuscript to eliminate presentation repetition between TokenPacker and our paper. The specific modifications include the second and third paragraphs of the Introduction, the Related Work section, the dataset section, and the normal resolution comparison in the Experiments section.
>
> **Q2:In lines 245-251 on page 5, the feature injection using enriched features through cross-attention shares a similar idea with Mini-Genimi and TokenPacker, although you employ deformable attention in this work.**
>
> REPLIES: Our approach differs from Mini-Gemini and TokenPacker in the following aspects: 1) The selection of injected tokens is different. Both Mini-Gemini and Tokenpacker use the downsampling operation to select the injected tokens, but our approach uses text features as the guidance to select the injected tokens, which can better align the text tokens and injected tokens. Here the injected tokens are the visual tokens fed into LLM. 2) Mini-Gemini and Tokenpacker use attention module to integrate the window features into the injected tokens, but this will cause the loss of valid information in other regions, while our method uses the deformable attention to fuse representative features from global features into the injected tokens.
>
> **Q3:Minor issue: In Table 1, line 344, the value 53.9 should be in bold with the best performance, instead of 53.5.**
>
> REPLIES: Thanks to the reviewer for pointing it out. We have revised this typo.
>
> **Q4:Considering the impressive performance, will the code be released publicly to facilitate reproduction of the results?**
>
> REPLIES: We fully agree with the reviewer's comments. We have provided the code in the supplementary material. After the anonymous review, we plan to open-source the code to GitHub.

---

> > ### Author Response · Authors · 2024-11-25
> > **Looking forward to the feedback on our responses**
> >
> > Dear Reviewer oteR
> >
> > We would like to express our sincere thanks again for the time and effort you spent on our paper. Since the discussion deadline is approaching, we kindly look forward to your feedback on our responses. We would be happy to discuss this further if there are still some open concerns.
> >
> > Best regards
> >
> > Authors

---

> > > ### Comment · Reviewer_oteR · 2024-11-26
> > >
> > > Thank you for the authors' response, my main concerns still persist:
> > >
> > > 1. (**Q1**) As the authors acknowledge, the initial version of the paper contained significant overlap with TokenPacker, including the Introduction, Related Work, dataset sections, and the comparisons presented in the Experiments section. This level of similarity is unacceptable for an academic paper.
> > > While the authors have made revisions, the overall presentation remains some similarities with  TokenPacker, including the conclusion section and the main tables.
> > > This issue was also highlighted by other reviewer.
> > >
> > > 2. (**Q2**) The explanations provided are superficial. The core idea remains the same: enhanced features are regarded as keys and values, while coarse features serving as the queries. Attention mechanisms are utilized to inject the enhanced key/values into the coarse queries. **The only difference is that you employ DeformAtt, whereas the previous work utilized the original attention mechanism.**
> > >
> > > I still believe this work is not well-prepared for publication due to its overall similar presentation and rough figures.
> > > Although this work achieves better performance, the contribution of the method appears to primarily stem from the combination of introducing language-guided operations from Grounding DINO and borrowing similar operations from TokenPacker.
> > >
> > > Therefore, I remain a negative recommendation regarding this paper.

---

> ### Author Response · Authors · 2024-11-26
> **Further discussion with Reviewer oteR**
>
> **Q1:Concerns about the presentation**
>
> REPLIES: Thanks for your reply. In the latest revision, we have carefully revised the full text, including the abstract, main results, and conclusion. Besides, we have also revised the style of the main table. In this revision, we have tried our best to improve the presentation and believe that the presentation is different from that of TokenPacker. Hope this can solve your concerns. If you still think there exists the presentation similarity, please **point them out**. We will try our best to revise them.
>
> **Q2:Concerns about the difference between Tokenpack, Mini-Gemini, and our LVP**
>
> REPLIES: Here we would like to clarify this question further. First, **the core innovation of our approach is to use textual features to filter important visual tokens**, as demonstrated by the visualization results in our paper. Selecting the visual tokens aligned with the text tokens can effectively improve the performance of the model. TokenPacker and Mini-Gemini only use downsampling to decide the injected tokens, which is under-performing. **We hope that our explanation of the core innovations can further resolve your confusion. Our method does not differ only at DeformAtt**. Second, Mini-Gemini and Tokenpacker use an attention module to perform feature injection, leading to the loss of valid information in other regions. while our method uses the deformable attention to perform feature injection.
>
> **Q3:Concerns about the novelty**
>
> We disagree with the reviewer with high respect about the **the contribution of the method appears to primarily stem from the combination of introducing language-guided operations from Grounding DINO and borrowing similar operations from TokenPacker**.
>
> Here we would like to further clarify our innovation.
>
> **Motivation and method**. In the motivation of our method, we found that the visual token outputted from the visual projector is not aligned with the textual token, **which has not been discussed in previous work**. Based on this finding, the direct way is using the textual features to select the visual tokens since the number of visual tokens is much larger than textual tokens. We depend on this and make the idea work and hope that this idea can be an effective trick for future design of visual projectors.  The motivation and methods are acknowledged by Reviewer goFW and sWSo.
>
> **Difference**. In order to highlight the difference with the existing methods, we have compared Grounding-DINO with our method in the initial submission. Our method is a plug-and-play module but the token selection in Grounding-DINO highly depends on its whole architecture. Besides, the selection of visual tokens in Grounding-DINO is a smaller part of the contribution, and its overall performance relies on the heavy decoder and the visual encoder. We also conducted experiments to demonstrate that such a heavy architecture does not achieve better performance.
>
> For the differences between TokenPacker and our LVP, we have elaborated above, i.e., textual features to select important visual tokens as well as DeformableAtt.
>
> Therefore, our method is not **a combination of introducing language-guided operations from Grounding DINO and borrowing similar operations from TokenPacker** but an effective visual projector, aligning text tokens and visual tokens.
>
> **Author's Claims**. Based on the fact that we have fully discussed on difference between Grounding-DIINO and our LVP in the initial submission. **We think taking this as the reason for the negative recommendation is inappropriate in the current stage, i.e., the end of rebuttal**. Hope the above discussion can help you understand our novelty and give our paper a chance to reconsider it.
>
> We would like to express our sincere thanks to the reviewers again.

---

> > ### Author Response · Authors · 2024-12-02
> > **Looking forward to the feedback**
> >
> > Dear Reviewer oteR
> >
> > We would like to express our sincere thanks for the time and effort you spent on our paper. Since the discussion deadline is approaching, we kindly look forward to your feedback on our latest responses.
> >
> > Best regards
> >
> > Authors

---

### Official Review · Reviewer_sWSo · 2024-11-03

**Soundness:** 3
**Presentation:** 3
**Contribution:** 2
**Rating:** 6
**Confidence:** 4

**Summary:**

This paper discusses advances in multimodal large language modeling by presenting a novel language-guided visual projector (LVP).The LVP utilizes textual features to guide the selection of visual tokens, which enhances the consistency between visual and textual data and improves computational efficiency. It employs a cross-modal feature enhancement module and a multi-level approach to capture fine-grained features, reducing the number of visual markers while maintaining performance. Experiments show that LVP achieves state-of-the-art results using fewer visual markers, demonstrating its effectiveness in multimodal tasks. Its main contributions include the innovative use of linguistic knowledge to optimize visual marker selection and improve multimodal learning.

**Strengths:**

1) The paper introduces a new approach to align visual and text tokens by using language features to guide the selection of essential visual tokens, which is novel in reducing token input for LLMs.
2) LVP improves the alignment between text and visual features with a cross-modal enhancement module, facilitating better integration of multimodal data.
3) By reducing visual tokens to 25% of the original input, LVP maintains high performance while enhancing computational efficiency in text generation tasks.
4) The multi-level projector design captures both detailed and global visual features, enhancing the system’s versatility and adaptability across different tasks.

**Weaknesses:**

1) The generalization of the method mentioned in the paper in the scenario of multi-round conversations with multi-modal large models is yet to be proved.
2) The structure proposed in the paper is similar to the one mentioned in paper LXMERT: Learning Cross-Modality Encoder Representations from Transformers, with an additional step of visual token selection.

**Questions:**

1) It is suggested to add relevant experiments to show that LVP still has excellent performance in the scenario of multi-round conversations with large multimodal models;
2) Specify the innovativeness of the structure, and make a distinction and appropriate citation from the more popular structures previously developed.

---

> ### Author Response · Authors · 2024-11-22
> **Response to Reviewer sWSo**
>
> **Q1:It is suggested to add relevant experiments to show that LVP still has excellent performance in the scenario of multi-round conversations with large multimodal models.**
>
> REPLIES: We appreciate the reviewer’s expertise and insight. **We have supplemented the multi-round conversations experiments on the MMDU dataset in Table 3 and Section 4.3 (Multi-round conversations)**. Following MMDU, we adopt Creativity (C), Richness (R), Visual Perception (VP), Logical Coherence (LC), Answer Accuracy (AA), and Image Relationship Understanding (IRU) as the evaluation metric.
>
> |Models|Param| C | R | VP | LC | AA | IRU |Avg.|
> |----|----| ---- | ---- | ----| ----| ----| ----|----|
> |LLaVA1.5-7B| 7B | 27.8 | 28.0 | 33.2| 43.0| 35.4| 31.7|32.2|
> |Qwen-VL-7B| 7B | 33.4 | 33.6 | 39.2| 53.8| 43.1| 38.1|39.3|
> |InternLM-XC2| 7B | 29.7 | 29.5 | 36.2 | 50.1 | 40.3 | 35.2 |35.6|
> |MiniCPM-v-2.5| 8B | 27.0 | 26.4 | 33.2 | 48.9 | 38.6 | 32.2 |33.0|
> |Deepseek-VL| 8B | 27.3 | 27.7 | 31.2 | 38.7 | 33.2 | 30.0 |30.8|
> |InternVL-Chat-v1.5| 26B | 31.2 | 31.5 | 37.4 | 52.6 | 41.7 | 36.1 |37.4|
> |LLaVA1.5+MMDU-45k| 7B | 34.3 | 34.5 | 36.7 | 47.2 | 38.5 | 35.5 |37.2|
> |LLaVA1.5-LVP+MMDU-45k| 7B |$\bf34.7$ | $\bf 35.0$ | $\bf 37.8$ | $\bf 49.0$ | $\bf 40.0$ | $\bf 36.0$ |$\bf 38.8$|
> |InternLM-XC2+MMDU-45k| 7B | 45.6 | 43.9 | 49.9 | 64.1 | 53.0 | 48.7 |50.1|
> |InternLM-XC2-LVP+MMDU-45k| 7B |$\bf 46.0$ | $\bf 44.4$ | $\bf 51.0$ | $\bf 65.7$ | $\bf 53.8$ | $\bf 49.0$ |$\bf 51.7$|\
>
> We can observe that InternLM-XC2-LVP establishes the new state-of-the-art results on each metric. LVP gains 1.6% improvement for LLaVA1.5 (38.8% vs. 37.2%) and InternLM-XC2 (51.7% vs. 50.1%). LVP improves performance by over 1% in terms of visual perception and logical coherence. The results demonstrate that LVP works for multi-round conversations.
>
> **Q2:Specify the innovativeness of the structure, and make a distinction and appropriate citation from the more popular structures previously developed.**
>
> REPLIES: Here we would like to make further clarification. The innovation of our structure lies in two ways: **Language-guide visual token selection** and **cross-modal feature enhancement module**. Language-guide visual token selection is the core innovation, which adopts the text feature as the guidance to select the important visual tokens, effectively aligning the visual tokens and text tokens. To our knowledge, adopting the text feature to decide the visual tokens fed into LLM is the first time.
>
> The cross-modal feature enhancement module differs from the existing method like LXMERT in the following aspects: 1) Image-to-text attention and Text-to-image attention in LXMERT are parallel, while our method is a sequential structure. 2) The cross-modality encoder in the LXMERT structure is much heavier than our cross-modal feature enhancement module. We conduct the experiment to show that LXMERT is not much superior to our method, demonstrating that such a lightweight module is enough for our method. Results are shown in the Table below.
>
> |Models|TPS|VQA$^\text{v2}$ |GQA |VQA$^\text{T}$ |OCRB |
> |----|----|----|----|----|----|
> |LXMERT|20.2|$\bf78.7$|61.8|57.6|$\bf299$|
> |LVP|$\bf25.3$|78.5|$\bf62.0$|$\bf58.0$|298|
> |LXMERT$^\sharp$|15.1|79.0|$\bf 62.6$|58.8|$\bf319$|
> |LVP$^\sharp$|$\bf24.2$|$\bf79.2$|62.5|$\bf58.9$|317|
>
> $^\sharp$ represents the multi-level language-guide visual projector. **We have made a distinction and citation from LXMERT in Line 213-238 in section 3.3. We also add the comparison experiment in Table A4 in the appendix**.

---

> > ### Author Response · Authors · 2024-11-25
> > **Looking forward to the feedback on our responses**
> >
> > Dear Reviewer sWSo
> >
> > We would like to express our sincere thanks again for the time and effort you spent on our paper. Since the discussion deadline is approaching, we kindly look forward to your feedback on our responses. We would be happy to discuss this further if there are still some open concerns.
> >
> > Best regards
> >
> > Authors

---

> > ### Comment · Reviewer_sWSo · 2024-11-25
> >
> > Glad to see your positive additional experiments and answers, I think this paper slightly exceeds the acceptance criteria

---

> > > ### Author Response · Authors · 2024-11-25
> > > **Many thanks to Reviewer sWSo**
> > >
> > > We would like to express our sincere thanks to the reviewer for the insightful feedback and support! Your suggestions are beneficial in improving the quality of our paper.

---

### Author Response · Authors · 2024-11-22
**Sincerely thanks to all reviewers**

We would like to express our sincere thanks to the reviewers for their insightful comments and helpful suggestions for the improvement of our paper. We have made a thorough modification, addressing the concerns of the reviewers one by one in this revision. Please note that we have highlighted the modified places in blue in the revision.

---

### Meta-Review · Area_Chair_taf1 · 2024-12-16

**Metareview:**

This paper proposes a new language-guided visual projector for large multimodal models. Reviewers raised significant concerns regarding the limited novelty compared to Grounding Dino and TokenPacker, the lack of sufficient details, and issues with the paper’s presentation.

In the rebuttal, the authors introduced many new experiments, conclusions, and substantial manuscript revisions, which addressed some of these issues. However, during the post-rebuttal phase, reviewers maintained the concern about the novelty and found relevant details are missing such as the token settings for high-resolution methods. Moreover, the extensive revisions provided during the rebuttal phase are not encouraged.

Given these considerations, the AC recommends rejection. The authors are encouraged to incorporate the reviewers' comments and re-submit to a future venue.

**Additional Comments On Reviewer Discussion:**

- Reviewer oteR, hBbQ and goFW agreed that the paper has limited novelty compared to Grounding Dino and TokenPacker though reviewer goFW considered the good performance improvement overweights the novelty issue.

- Reviewer oteR and hBbQ found the paper still lacks sufficient details even after the rebuttal, particularly regarding the $N_a$ setting in multi-level design and its relationship with the original $N_q$ in Eq (2), as well as the high-resolution implemtentations. There seems to be problems in the tables and the token settings in high-resolution method.

- The original paper closely resembles TokenPacker in presentation and the figures should be polished (reviewers oteR and hBbQ). Though the authors revised the paper in the rebuttal, the extent of changes raises concerns about whether such significant modifications should be made post-submission.

---

### Decision · Program_Chairs · 2025-01-22

Reject